# Scaling RL to Long Videos

Yukang Chen[1*]    Wei Huang[1,3*]    Baifeng Shi[1,4†]    Qinghao Hu[2†]    Hanrong Ye[1†]

Ligeng Zhu[1]    Zhijian Liu[1]    Pavlo Molchanov[1]    Jan Kautz[1]    Xiaojuan Qi[3]

Sifei Liu[1]    Hongxu Yin[1]    Yao Lu[1]    Song Han[1,2]

[1]NVIDIA    [2]MIT    [3]HKU    [4]UC Berkeley

## Abstract

We introduce a full-stack framework that scales up reasoning in vision-language models (VLMs) to long videos, leveraging reinforcement learning. We address the unique challenges of long video reasoning by integrating three critical components: (1) a large-scale dataset, LongVideo-Reason, comprising 104K long video QA pairs with high-quality reasoning annotations across diverse domains such as sports, games, and vlogs; (2) a two-stage training pipeline that extends VLMs with chain-of-thought supervised fine-tuning (CoT-SFT) and reinforcement learning (RL); and (3) a training infrastructure for long video RL, named Multi-modal Reinforcement Sequence Parallelism (MR-SP), which incorporates sequence parallelism and a vLLM-based engine tailored for long video, using cached video embeddings for efficient rollout and prefilling. In our experiments, LongVILA-R1-7B achieves strong performance on video benchmarks, reaching 65.1% and 71.1% accuracy on VideoMME without and with subtitles, respectively, and consistently outperforming LongVILA-7B across multiple benchmarks. Moreover, LongVILA-R1-7B supports processing up to 8,192 video frames per video, and configurable FPS settings. Notably, our MR-SP system achieves up to $2.1\times$ speedup on long video RL training. In addition, we release our training system for public availability that supports RL training on various modalities (video, text, and audio), various models (VILA and Qwen series), and even image and video generation models. On a single A100 node (8 GPUs), it supports RL training on hour-long videos (*e.g.*, 3,600 frames). Code and models are available at https://github.com/NVlabs/Long-RL and https://huggingface.co/Efficient-Large-Model/LongVILA-R1-7B.

## 1 Introduction

Understanding long videos requires more than simple recognition—it demands reasoning from temporal, spatial, goal-oriented, and narrative perspectives [51]. As illustrated in Figure 1, answering high-level questions often hinges on a model's ability to integrate clues distributed across time, infer hidden goals or strategies, track entities spatially, and comprehend the evolving plot. For instance, predicting the winner of a football penalty shootout involves assessing emotional cues and tactical behavior (temporal and goal reasoning), while determining the final location of a hidden ball requires precise spatial tracking. Likewise, evaluating a poker player's decision demands interpreting implicit strategies beyond surface actions (goal reasoning) and understanding a character's development or match trajectory reflects the need for plot reasoning. These examples underscore that reasoning is indispensable for long video understanding that goes beyond recognition alone. Despite the clear importance of reasoning in long video understanding, enabling such capabilities in long video VLMs poses significant challenges [8, 43, 47, 12, 37]. First, the collection of high quality long video reasoning datasets is inherently difficult. Unlike domains such as math or code reasoning, where

---

*Equal contribution    †Core contribution

39th Conference on Neural Information Processing Systems (NeurIPS 2025).

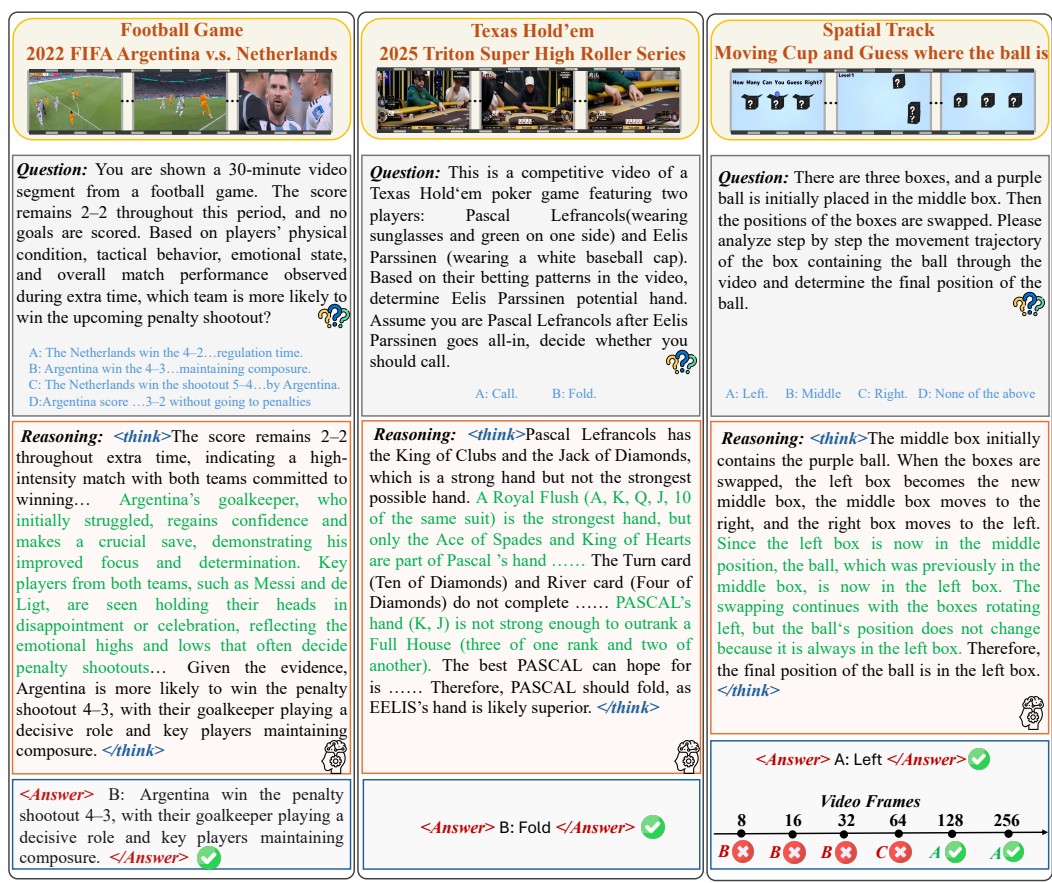

Figure 1: Examples of LongVILA-R1. The illustration demonstrates sample tasks and their reasoning. From left to right, the examples include predicting the results of a football match, decision-making reasoning in Texas Hold'em Poker, and trajectory for spatial dynamics of objects. Notably, the spatial tracking video involves a relatively complex dynamic moving, for which the model fails to achieve accurate reasoning until the number of input video frames increases to 128.

structured supervision and benchmarks are readily available [35, 24], long video reasoning requires annotating complex temporal dynamics, goals, spatial relations, and narrative elements—often across minutes or hours of footage [14]. This process is labor-intensive and subjective, making large-scale dataset construction slow and costly. Second, the RL training framework for long videos is challenging. Reinforcement learning, a common strategy for aligning models with complex reasoning objectives, is computationally expensive and sample-inefficient [38]. When applied to long videos, RL becomes even more burdensome due to the extended video frames, requiring more memory and longer rollout runtime. These challenges jointly hinder the development of effective long video VLMs with strong reasoning capabilities.

In this work, we introduce LongVILA-R1, a comprehensive framework exploring the reasoning capabilities for long video understandings. Firstly, we strategically construct a high quality dataset with CoT annotations for long video reasoning, named LongVideo-Reason. Leveraging a powerful VLM (NVILA-8B) [27] and a leading open-source reasoning LLM, we develop a dataset comprising 104K high quality *Question-Reasoning-Answer* pairs for long videos. We use 36K high quality samples for Long-CoT-SFT to initialize the model's reasoning and instruction-following abilities, and 68K samples with an additional 102K video data [53, 46, 31, 18, 44] for reinforcement learning. This two-stage training combines high quality reasoning annotations with reinforcement learning, enabling LongVILA-R1 to achieve superior and generalized video reasoning. We also manually curate a balanced set of 1K long video samples to build a new benchmark, LongVideo-Reason-eval, that evaluates performance from four perspectives: *Temporal*, *Goal and Purpose*, *Spatial*, and *Plot and Narrative*, for a comprehensive assessment.

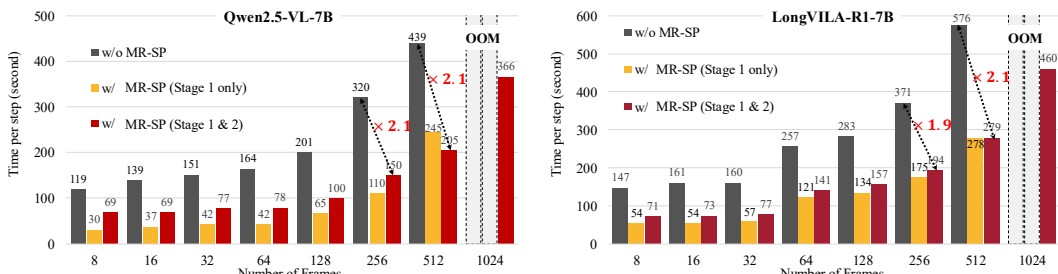

Figure 2: Training efficiency comparison with MR-SP (SP degree=4) on Qwen2.5-VL-7B and LongVILA-R1-7B and a single node 8× A100 GPUs. It achieves 2.1× speed-up and avoids GPU OOM issue on long frames.

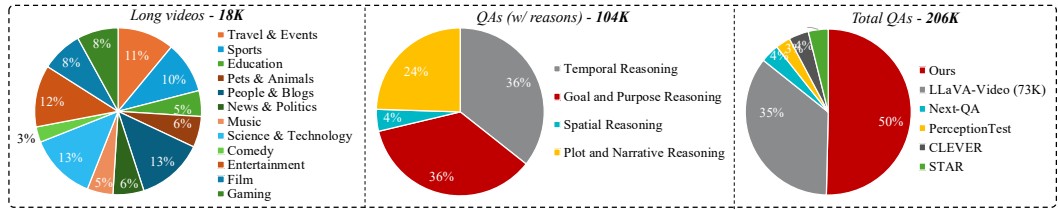

Figure 3: Data Distribution of LongVideo-Reason and total data in the LongVILA-R1 training framework. LongVideo-Reason comprises a total of 18K videos and 104K QAs with reasoning annotations. Additionally, we include 102K QAs from existing works [53, 46, 31, 18, 44].

Secondly, we propose a training framework for VLMs to advance long video reasoning. As illustrated in Figure 5, this framework incorporates two stages, *i.e.*, Stage-1: Long CoT-SFT, and Stage-2: RL for long video reasoning. To address the unique challenges of long video RL, including massive visual embeddings, heavy rollouts, and long-context LLM prefilling, we develop an efficient and scalable solution, referred to as Multi-modal Reinforcement Sequence Parallelism (MR-SP). It incorporates a vLLM engine [19] tailored for LongVILA and a caching scheme for video embeddings. The MR-SP system alleviates the problem of intensive memory and facilitates RL training of long video VLMs. As shown in Figure 2, our MR-SP system achieves up to 2.1× speedup on 512-frame video RL training on 7B models and enables longer training frames without out-of-memory (OOM).

In our experiments, LongVILA-R1-7B demonstrates strong performance on video benchmarks, achieving 65.1% and 71.1% accuracy on VideoMME without and with subtitles, respectively. It consistently outperforms LongVILA-7B across a range of benchmarks, including ActivityNet-QA, LongVideoBench, PerceptionTest, NExT-QA, VNBench, and VideoMME. Additionally, LongVILA-R1 supports processing up to 8,192 video frames per video, and configurable FPS settings. On our LongVideo-Reason-eval benchmark, LongVILA-R1-7B achieves an average accuracy of 72.0%, surpassing Video-R1-7B[8], as well as proprietary models such as Gemini-1.5-Pro[34].

## 2 Related Work

**Multi-modal reasoning models.** The field of multi-modal reasoning has advanced significantly, particularly in Vision-Language Models (VLMs). GPT-4o [29] improves visual understanding through enhanced reasoning, while Gemini-1.5-Pro [34] extends context length to 1 million tokens, achieving state-of-the-art performance in VideoMME [9]. Following the substantial progress of architecture and training algorithms in open-source VLMs [56, 27, 48, 23, 39], multi-modal reasoning has been further explored in works including LMM-R1 [32] which employs a two-stage training strategy, Vision-R1 [15] that addresses post-cold-start overthinking, and Video-R1 [8] which enhances RL for video via T-GRPO in 16 frames. However, these approaches primarily focus on single images or short videos, and long video reasoning still poses great challenges.

**Sequence parallelism.** Training with long contexts often exceeds the memory capacity of a single device [4], necessitating efficient distribution strategies. Sequence parallelism (SP) has become a

widely adopted solution [25, 16, 3, 7, 11]. For example, ring-based systems like LightSeq [20] and Ring Attention [25] use point-to-point (P2P) communication, while DeepSpeed-Ulysses [16] employs all-to-all (A2A) primitives to optimize attention computations. Additionally, USP and LoongTrain [7, 11] were introduced to integrate Ring-style SP and Ulysses SP. LongVILA [47] further proposed multi-modal SP (MM-SP), enabling vision-language models to handle long-context inputs. However, multi-modal reinforcement learning introduces additional challenges, as it requires extensive sampling from long, mixed-token sequences [38], particularly in complex group optimization tasks [35].

**RL frameworks for LLMs/VLMs.** Reinforcement Learning (RL) has become a key strategy for enhancing Large Language Models (LLMs), particularly through Reinforcement Learning with Human Feedback (RLHF) [30] or Direct Preference Optimization [33], which aligns model outputs with human preferences. Recent advancements demonstrate that RL significantly improves LLM reasoning abilities. For example, DeepSeek-R1 [6] utilizes the Group Relative Policy Optimization (GRPO) algorithm [36], integrating group-based sampling and rule-based rewards. On the other hand, RL [1] poses a unique challenge of heavy computational cost, especially in multi-modal settings. To address this problem, HybridFlow [38] is introduced, leveraging Ray [28] for efficient data flow and vLLM [19] for faster sampling. Nevertheless, this remains a bottleneck when processing long video sequences, with group-based sampling constrained by the high computational cost of long-context sampling and visual encoding. In this work, we propose MR-SP that provides up to $2.1\times$ speedup.

# 3 LongVideo-Reason Data Construction

## 3.1 Overview of Data Curation

We first curate 18K long videos from the Shot2Story dataset [13] (Figure 3, left). We further incorporated 2k additional 4K-resolution videos spanning scenarios such as autonomous driving, video games, household robotics, and wildlife. We then apply a high quality automated annotation pipeline for CoT as detailed in Section 3.2, and end up with a total of 104K Question-Reasoning-Answer pairs where each sample, based on the type of question it is reasoning about, can be categorized into *Temporal Reasoning*, *Goal and Purpose Reasoning*, *Spatial Reasoning*, or *Plot and Narrative Reasoning* (Figure 3, middle). This dataset is designed to support various types of long-video reasoning tasks comprehensively.

Given the sensitivity of GRPO to batch sampling [40, 32], a data filtering approach is adopted. Specifically, we employ a test-scaling method where LongVILA [47] performs inference 10 times on the original datasets. Questions consistently answered correctly or incorrectly are labeled as easy or hard while those inducing diverse predictions are labeled as medium. We filter out too easy and too hard questions, and use remaining questions for training. The reason is that GRPO expects different rollouts of each sample to be diverse in order to have meaningful advantages, and the gradient vanishes if all the rollouts predict correct or incorrect answers. The COT-SFT subset (36K) features high quality CoT reasoning processes formatted in a standard *<think></think><answer></answer>* structure, providing abundant resources for warm-up training during Stage 1 of the model's reasoning capabilities. Meanwhile, the RL subset contains 68K challenging long-video Q&A, which is leveraged in Stage 2 for scaling reasoning through reinforcement learning. To further enhance RL scaling, we incorporate an additional 102K high quality open-source videos (Figure 3, right) from other datasets [53, 46, 31, 18, 44]. This combination improves the model's generalization.

## 3.2 Long-Video Reasoning Generation

We introduce an automated annotation pipeline (Figure 4) that generates high quality *Question-Reasoning-Answer* pairs from long videos. This process begins by segmenting videos into short clips (~10 seconds each), each of which is annotated using the NVILA-8B [27] model to provide descriptive captions. For the spatial reasoning category, We employed VILA-HD [39] to generate object bounding boxes in video frames and constructed spatial reasoning QAs based on these boxes and corresponding captions. Leveraging the breakthrough in text-based reasoning, we then deploy a leading open-source reasoning LLM, DeepSeek-R1-671B [6], provide the captions of all the clips in each video, and prompt it to generate diverse types of *Question-Reasoning-Answer* pairs that involve reasoning over the content across the whole video.

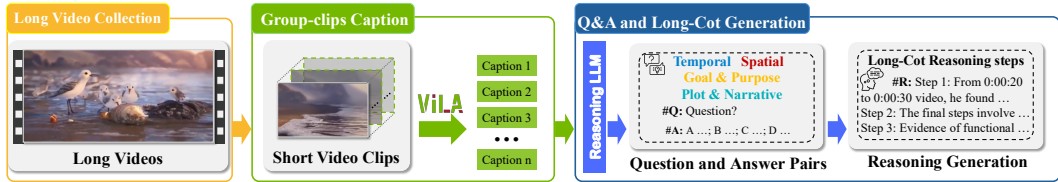

Figure 4: Data generation process for the LongVideo-Reason dataset. This process begins with segmenting videos into 10-second clips and generating captions for each clip using NVILA-8B. Then based on the captions of all clips in a video, we generate question-answer pairs that involve reasoning across the content of the whole video, along with the reasoning annotations using a leading open-source reasoning LLM. Reasoning questions are categorized into Temporal, Goal and Purpose, Spatial, and Plot and Narrative. Finally, the reasoning annotations are reformatted for conciseness and alignment with video details. We present a more detailed figure of data generation process in Figure 9.

Specifically, we design four types of prompts to encourage the LLM to generate a *Question-Reasoning-Answer* pair that focuses on one of the four types of reasoning: *Temporal Reasoning*, *Goal and Purpose Reasoning*, *Spatial Reasoning*, or *Plot and Narrative Reasoning*. To ensure VLMs focus on visual details, we also craft the prompts with phrases such as *"checking the video"* and *"analyzing the scene"*, which guide iterative examination of visual content. Finally, an LLM is then used to refine and streamline the reasoning steps. We also manually curated 1,000 high quality complex reasoning questions across four reasoning categories to serve as a new benchmark (LongVideo-Reason-eval) for evaluating VLMs in reasoning abilities. Note that we also include open-ended questions. Among the total 104K QA pairs, approximately half are multiple-choice and hald are open-ended. This entire data procedure consumes about 80,000 H100 GPU hours.

## 4 LongVILA Training Pipeline

As shown in Figure 5, there are two extended training stages in LongVILA-R1, *i.e.*, (1) warm-up for long video reasoning, utilizing 36K data with high quality CoT for SFT on the MM-SP system [47]; (2) reinforcement learning with dense frames from long videos.

### 4.1 Long Video CoT Supervised Fine-Tuning

Utilizing 104K high quality question-reasoning-answer pairs, we apply the data filtering method described in Section 3.1 to select 36K examples for long CoT-SFT, serving as a warm-up phase for subsequent RL. This stage equips the model with fundamental reasoning abilities and instruction-following skills for long video scenarios. To efficiently perform SFT on hundreds of frames, we adopt the MM-SP [47] training system from LongVILA. As demonstrated in Section 6.2, SFT solely on our LongVideo-Reason dataset also effectively improves the model with reasoning capabilities.

### 4.2 GRPO for Long Video

Building on the advancements of the GRPO [35] algorithm and prior explorations of multi-modal reasoning training [8, 32], we adhere to the standard GRPO framework to train our model. For each given question $q$, the policy model generates a group of candidate responses $\{o_1, o_2, ..., o_G\}$ from the old policy $\pi_{\theta_{old}}$, accompanied by their corresponding rewards $\{r_1, r_2, ..., r_G\}$, which are computed based on rule-based reward functions (format/accuracy). The model $\pi_\theta$ is subsequently optimized by maximizing the following objective function:

$$\mathcal{J}(\theta) = \mathbb{E}_{q,\{o_i\}}\big[\frac{1}{G}\sum_{i=1}^{G}(\min(\frac{\pi_\theta(o_i|q)}{\pi_{\theta_{old}}(o_i|q)}A_i,$$
$$\text{clip}(\frac{\pi_\theta(o_i|q)}{\pi_{\theta_{old}}(o_i|q)}, 1-\epsilon, 1+\epsilon)A_i) - \beta\mathbb{D}_{KL}(\pi_\theta||\pi_{ref}))\big] \quad (1)$$

where $\epsilon$ and $\beta$ are hyper-parameters, $G$ is set as 8 in our experiments, and the sampled rewards above are normalized to get the advantages ($A_i$) for updating the model:

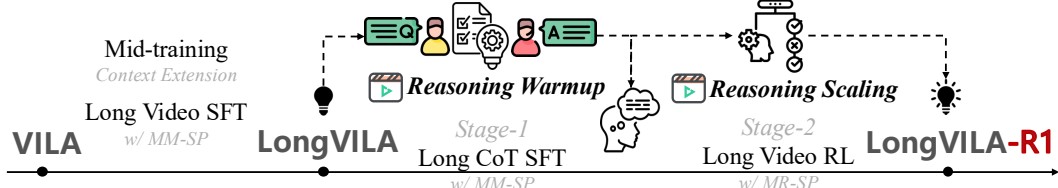

Figure 5: The LongVILA-R1 training pipeline. LongVILA-R1 builds upon the base training pipeline for LongVILA. MM-SP is further employed for SFT on long video understanding tasks with long CoT. Then, reinforcement scaling learning is conducted through Multi-modal Reinforcement Sequential Parallelism (MR-SP).

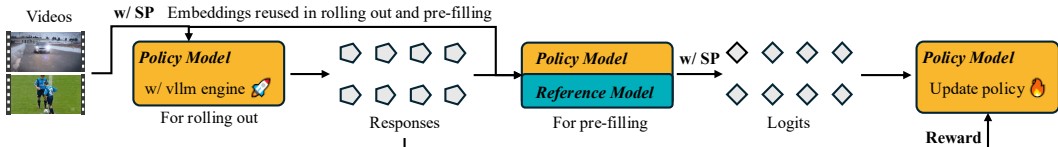

Figure 6: LongVILA-R1 RL training framework. The framework integrates multi-modal reinforcement sequence parallelism (MR-SP) for scalable video frame encoding and LLM prefilling. RL utilizes a vLLM-based engine with cached video embeddings, tailored for LongVILA rollout. Rewards for accuracy and format guide policy optimization.

$$A_i = \frac{r_i - \text{mean}(\{r_1, r_2, ..., r_G\})}{\text{std}(\{r_1, r_2, ..., r_G\})} \tag{2}$$

However, RL for long videos presents significant challenges due to the high computational demands of processing hundreds to thousands of frames. Existing RL frameworks struggle with such long-context training in rollout and LLM prefilling. To address this, we develop the MR-SP framework (Section 5), which efficiently scales reinforcement learning for long-context video reasoning.

Considering the sensitivity of GRPO to sampling during training [40], we use the 68K filtered data for reinforcement learning as described in Section 3.1. Additionally, an extra 102K samples from other datasets [53, 46, 31, 18, 44] are incorporated to scale up the RL. This approach aims to guide the model in freely exploring and developing more effective and generalized reasoning strategies.

## 5 Multi-modal Reinforcement SP

Existing RL frameworks for VLMs, such as R1-V [1] and EasyR1 [55], are not designed for long videos which present unique challenges due to their high token volume. To address this, we introduce Multi-modal Reinforcement Sequence Parallelism (MR-SP), a framework for efficient RL training on long videos. MR-SP leverages sequence parallelism in both rollout and pre-filling stages, enabling long videos in RL, with reduced overhead. We show the training curve with MR-SP in Figure 8.

### 5.1 Stage 1 - Rollout with Paralleled Encoding

To support long-video reinforcement learning efficiently, we adopt sequence parallelism (SP) for the video encoding stage. As shown in Figure 7, the input video frames are first evenly divided across multiple GPUs (e.g., GPU 1 to GPU 3), each equipped with its own vision tower. Each GPU independently processes a slice of the video, encoding only a subset of the frames. The resulting video embeddings are then aggregated with text embeddings via an all-gather operation as indicated by the "All-Gather" arrow in the figure. This strategy distributes the encoding workload, allowing the system to handle significantly longer videos by leveraging more GPUs, while avoiding the risk of GPU memory overflow. The parallel encoding scheme ensures balanced utilization of the vision towers and enables scalable long-video processing that would otherwise be infeasible.

After the video embeddings are globally gathered, they are **reused** for downstream usage throughout the RL pipeline. As illustrated in Figure 7, the gathered embeddings are reused during multiple

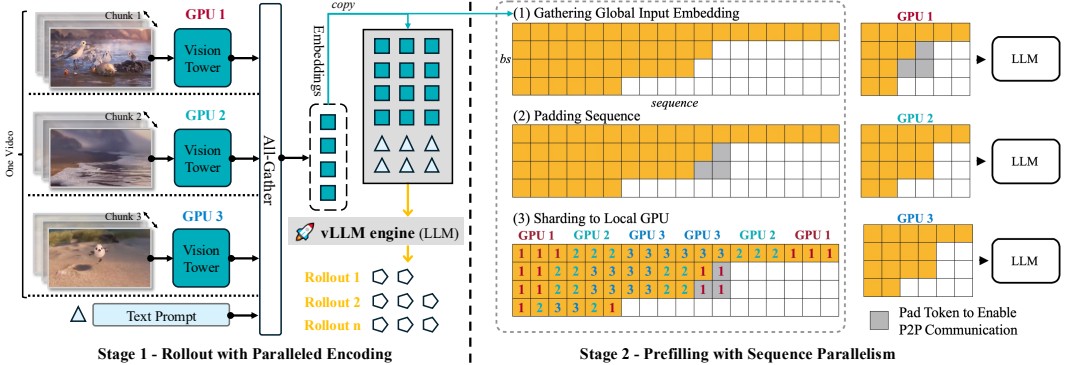

Figure 7: The workflow of multi-modal reinforcement sequential parallel (MR-SP). To accommodate multi-modal input in reinforcement learning, we develop a custom sharding strategy to ensure balanced workload distribution and compatibility with SP communication. Efficient video embedding reuse and vLLM rollout acceleration strategies are implemented, while meeting the demands of policy model prefilling for dense video frames.

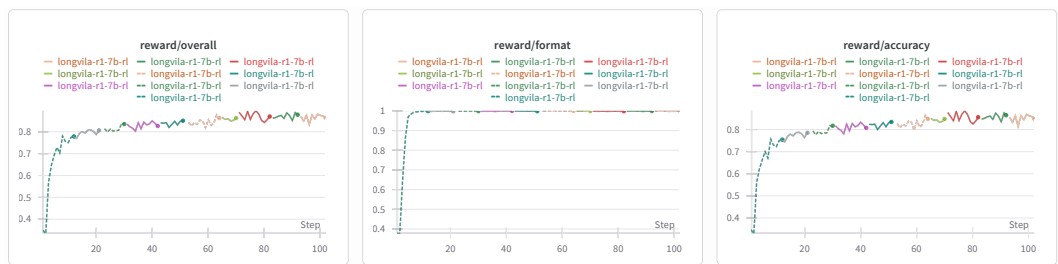

Figure 8: Reward curve of our LongVILA-R1-7B with MR-SP. During training, LongVILA-R1-7B exhibits stable reward improvements across overall, format, and accuracy dimensions.

rollouts without recomputation. For instance, in each training step, we typically perform 8 to 16 rollouts. Without recycling, the same video would need to be re-encoded dozens of times per step, severely impacting training speed. By caching and reusing the gathered embeddings, MR-SP eliminates this redundancy and significantly accelerates training.

### 5.2 Stage 2 - Prefilling with Sequence Parallelism

For each rollout, both the reference and policy models require prefilling compute-intensively in RL for long videos. With the gathered embedding from Stage 1 **reused**, we parallelize the inference stage across devices using sequence parallelism. As illustrated in Figure 7, we globally gathered input embeddings are first padded to a uniform length (Padding Sequence) and then evenly partitioned across GPUs (Sharding to Local GPU). This allows each GPU to handle only a portion of the input sequence during prefilling. This parallelism is applied to both policy and reference model prefilling. Then, each GPU locally computes logits for its token slice, prefilling in parallel.

## 6 Experimental Results

### 6.1 Main Results

Table 1 shows the performance comparison on 6 video benchmarks [49, 45, 31, 46, 54, 9]. LongVILA-R1-7B consistently outperforms LongVILA-7B across all benchmarks, with performance gaps that vary according to the complexity of the reasoning tasks. Table 3 presents the general performance of LongVILA-R1, comparing to existing advanced models [22, 52, 2, 21, 17, 26, 42, 51, 8, 57, 41, 5, 53, 27, 50] under comparable model sizes on the Video-MME [9] benchmark. The LongVILA-R1-7B is tested using 512 video frames as inputs. LongVILA-R1-7B achieves leading scores across different video lengths, obtaining scores of 65.1% and 71.1% in the settings without subtitles and with subtitles,

Table 1: Performance on ActivityNet-QA [49], LongVideoBench [45], PerceptionTest [31], NExT-QA [46], VNBench [54], and VideoMME [9]. LongVILA-R1-7B outperforms LongVILA-7B across all these benchmarks, with varying margins.

| Model | Act.Net-QA test | L.V.Bench val | Per.Test val | NExT-QA mc | VNBench val | VideoMME w/o sub. | VideoMME w/ sub. |
|---|---|---|---|---|---|---|---|
| GPT-4o mini | - | 56.5 | - | - | - | 64.8 | 68.9 |
| GPT-4o | 61.9 | 66.7 | - | - | 64.4 | 71.9 | 77.2 |
| Gemini-1.5-Pro | 57.5 | 64.0 | - | - | 66.7 | 75.0 | 81.3 |
| Video-LLaVA-7B | 45.3 | 37.6 | - | - | 12.4 | 39.9 | 41.6 |
| Flash-VStream-7B | 51.9 | - | - | 61.6 | - | - | - |
| ShareGPT4Video-8B | 50.8 | 41.8 | - | - | - | 39.9 | 43.6 |
| VideoLLaMA2-7B | 50.2 | - | 51.4 | - | 4.5 | 47.9 | 50.3 |
| VideoLLaMA2.1-7B | 53.0 | - | 54.9 | - | - | 54.9 | 56.4 |
| Kangaroo-8B | - | 54.8 | - | - | - | 56.0 | 57.6 |
| PLLaVA-7B | 56.3 | 39.2 | - | - | - | - | - |
| LLaVA-OV-7B | 56.7 | 56.4 | 57.1 | 79.4 | 51.8 | 58.2 | 61.5 |
| LongVILA-7B | 59.5 | 57.1 | 58.1 | 80.7 | 63.0 | 60.1 | 65.1 |
| LongVILA-R1-7B | **64.8** | **58.0** | **68.9** | **81.5** | **75.5** | **65.1** | **71.1** |

Table 2: Performance on LongVideo-Reason-eval. LongVILA-R1-7B achieves a strong overall score.

| Model | Temporal | Goal | Plot | Spatial | Overall |
|---|---|---|---|---|---|
| Video-R1-7B [8] | 61.4 | 85.0 | 62.0 | **58.5** | 68.1 |
| Gemini-1.5-Pro [34] | 65.4 | 81.9 | 67.8 | 53.3 | 69.3 |
| LongVILA-7B | 58.0 | 80.2 | 57.1 | 46.7 | 62.7 |
| LongVILA-R1-7B | **68.1** | **85.7** | **70.6** | 53.3 | **72.0** |

respectively. Table 2 compares the results of our LongVideo-Reason-eval benchmark. LongVILA-R1-7B model achieves a strong performance with an average score of 72.0%, surpassing Video-R1-7B [8] and slightly outperforms Gemini-1.5-Pro [34]. Qualitative results are in the appendix.

## 6.2 Ablation Study

**Scaling video frames.** The reasoning capability of LongVILA-R1 scales consistently with the number of input video frames. Specifically, Table 5 illustrates the performance of LongVILA-1.5B (grey line) and LongVILA-1.5B-R1 (red line) on the long-video reasoning benchmark under varying frame inputs. With only 16 input frames, LongVILA-R1-1.5B is close to LongVILA-1.5B. However, as the number of frames increases to 512, LongVILA-R1-1.5B consistently outperforms and eventually achieves a score of 64.3%. Notably, LongVILA-R1-1.5B demonstrates steady performance improvements throughout the scaling process. In contrast, LongVILA-1.5B hits a performance bottleneck with 256 frames, and got a degradation on 512 frames. The enhanced reasoning capabilities of LongVILA-R1-1.5B allow it to effectively integrate and infer information from long videos.

**Ablation on pipeline and datasets.** As shown in Table 4, we ablate the effectiveness of training stages and datasets, using LongVILA-1.5B as a starting point. The accuracies are evaluated on LongVideo-Reason-eval. ✗ means skipping this stage, ✓ means training this stage with our datasets, and **O** means training this stage with other datasets [53, 46, 31, 18, 44]. Our CoT-SFT dataset results in a better performance than that of other datasets. In addition, incorporating RL on top of the warm-up phase (CoT-SFT) yields additional improvements compared to using only SFT. We show that if we skip CoT-SFT and train our models directly with RL, the accuracy drops. If we apply Video-R1 datasets in both CoT-SFT and RL stages, the performance is inferior to using ours.

**Training efficiency on MR-SP.** We conduct the training efficiency comparison for our MR-SP system, one A100 node, *i.e.*, 8xA100 (80GB) GPUs. We measure the forward time for each training

Table 3: Performance comparison on VideoMME [9] benchmark in details.

| Model | w/o subtitle | | | | w subtitle | | | |
|---|---|---|---|---|---|---|---|---|
| | Overall | Short | Medium | Long | Overall | Short | Medium | Long |
| Video-R1-7B [8] | 61.4 | - | - | - | - | - | - | - |
| Apollo-7B [57] | 61.1 | - | - | - | 63.3 | - | - | - |
| LLaVA-Video-7B [53] | 63.3 | - | - | - | 69.7 | - | - | - |
| NVILA-8B-Video [27] | 64.2 | - | - | - | 70.0 | - | - | - |
| Video-LLaVA-7B [22] | 39.9 | 45.3 | 38.0 | 36.2 | 41.6 | 46.1 | 40.7 | 38.1 |
| SliME-8B [52] | 45.3 | 53.3 | 55.4 | 39.8 | 47.2 | 55.4 | 44.4 | 41.7 |
| ShareGPT4Video-8B [2] | 39.9 | 48.3 | 36.3 | 35.0 | 43.6 | 53.6 | 39.3 | 37.9 |
| VideoChat2-7B [21] | 39.5 | 48.3 | 37.0 | 33.2 | 43.8 | 52.8 | 39.4 | 39.2 |
| Chat-Univi-v1.5-7B [17] | 40.6 | 45.7 | 40.3 | 35.8 | 45.9 | 51.2 | 44.6 | 41.8 |
| Kangaroo-8B [26] | 56.0 | 66.1 | 55.3 | 46.7 | 57.6 | 68.0 | 55.4 | 49.3 |
| ShareGemini-7B [42] | 43.2 | 49.1 | 41.3 | 39.1 | 47.9 | 49.1 | 47.3 | 43.4 |
| LongVA-7B [51] | 52.6 | 61.1 | 50.4 | 46.2 | 54.3 | 61.1 | 53.6 | 47.6 |
| VITA-1.5-7B [10] | 56.1 | 67.0 | 54.2 | 47.1 | 58.7 | 69.9 | 55.7 | 50.4 |
| LongVILA-7B [47] | 60.1 | 69.0 | 58.3 | 53.0 | 65.1 | 72.9 | 64.9 | 57.4 |
| LongVILA-R1-7B | **65.1** | **76.8** | **63.2** | **55.2** | **71.1** | **79.2** | **69.7** | **64.3** |

Table 4: Ablations on frames and reasoning (R.).

| Frames | 16 | 32 | 64 | 128 | 256 | 512 |
|---|---|---|---|---|---|---|
| w/o R. | 55.7 | 56.4 | 58.1 | 60.5 | 60.7 | 60.2 |
| w/ R. | 55.9 | 56.7 | 61.9 | 62.6 | 64.1 | 64.3 |

Table 5: Ablations on CoT-SFT and RL.

| CoT-SFT | ✗ | ✓ | ✗ | O | O | ✓ |
|---|---|---|---|---|---|---|
| RL | ✗ | ✗ | ✓ | ✗ | O | ✓ |
| Accuracy | 58.1 | 60.2 | 52.4 | 59.1 | 59.4 | 61.9 |

step. The results are obtained after 10 warming up iterations and averaged over 5 iterations. We use LongVILA-7B-R1 model with training batch size as 1 per GPU and rollout number as 5.

Figure 2 presents the training efficiency comparison across different numbers of frames. The figure plots the runtime per step (in seconds) for three settings: the plain RL system without MR-SP, MR-SP Stage 1 only, and the full MR-SP system (Stage 1 & 2). The baseline runtime increases steeply as the frame count grows. Using only Stage 1 of MR-SP significantly improves efficiency up to 512 frames but encounters GPU out-of-memory (OOM) issues beyond that point. In contrast, the full MR-SP system consistently reduces runtime, achieving up to a 2.1× speedup at 512 frames and scaling efficiently to 1024 frames without OOM, highlighting the benefit of combining sequence reuse and sequence parallelism for long video RL training.

## 7 Conclusion

We present a comprehensive framework designed to fully scale VLMs for reasoning over long videos. LongVILA-R1 encompasses a meticulously constructed large-scale dataset, LongVideo-Reason, and a parallelized training framework, MR-SP. Leveraging our curated dataset of 104K long video question-reasoning-answer pairs, combined with other open-source video datasets, we adopt a two-stage training process that integrates CoT-SFT and RL. LongVILA-R1-7B demonstrates outstanding performance on mainstream video benchmarks, achieving 65.1% and 71.1% on VideoMME without and with subtitles. LongVILA-R1 supports processing up to 8,192 video frames per video, and configurable FPS settings. Notably, our MR-SP leads to a speed up of 2.1× for long video RL training, supporting hour-level (3,600 frames) RL training on a single node of 8 A100 GPUs. In addition, we release our training system to the public, which supports RL training across multiple modalities (video, text, and audio), various models (including VILA and Qwen series), and even image and video generation models.

**Limitations** While our RL training system is capable of handling long video frames efficiently (3,600 frames on a single node of A100 GPUs), scaling to significantly longer sequences, more modalities (like include audio for omni VLMs) or large batch sizes would require distributed training across multiple GPUs. This requires more GPUs, making it less feasible to run on limited resources.

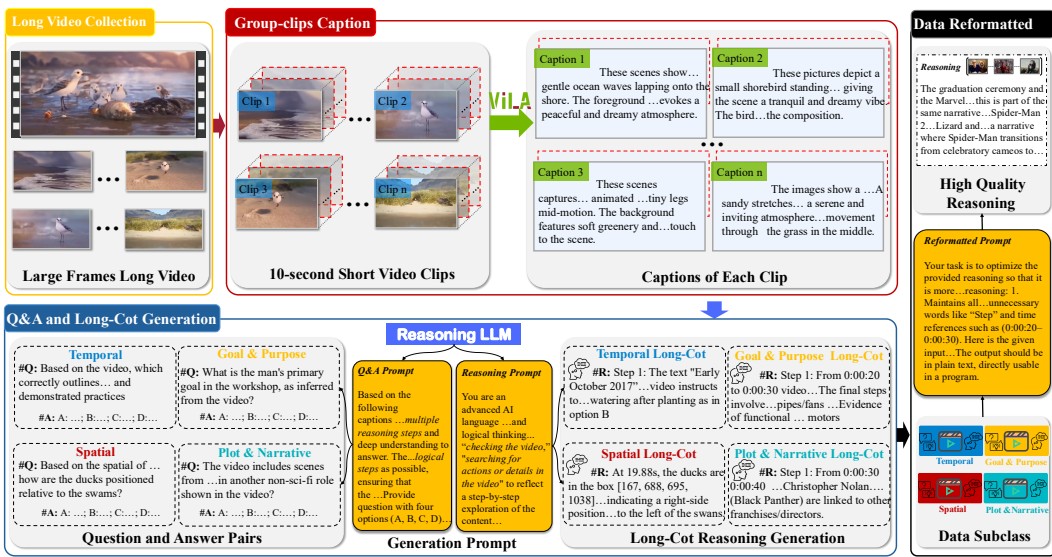

Figure 9: Detailed data generation process for the LongVideo-Reason dataset, supplementing Figure 4.

# 8 Border Impacts

The development of LongVILA-R1 represents a significant advancement in long video reasoning, enabling real world systems to perform sophisticated temporal and compositional understanding across diverse and extended visual contexts. This progress holds transformative potential across multiple domains, including embodied AI, robotics, autonomous systems, and AR/VR applications. By equipping vision-language models (VLMs) with the ability to process and reason over long-duration video data, LongVILA-R1 lays the foundation for AI systems capable of understanding event sequences, and inferring causal and physical relationships over extended frames.

Long video reasoning technologies powered by LongVILA-R1 can significantly enhance embodied AI and robotics by enabling agents to sustain coherent, long-term understanding of their environment. Robots equipped with such capabilities would excel in performing complex, multi-stage tasks, adapting to dynamic contexts, and building richer world models for planning and decision-making. These advancements also promise to unlock new opportunities in education, healthcare, and entertainment. For instance, long video understanding could enable AI tutors to analyze and summarize extended instructional videos, or assist healthcare professionals in reviewing lengthy procedural recordings. Furthermore, such systems could enhance sports analytics, and other areas requiring nuanced temporal reasoning.

Figure 9 shows the the data-generation where our supervision is built from *grouped short clips* that form long videos, per-clip *captions*, and LLM-generated *Q&A with Long-CoT* that is formatted as multiple-choice reasoning rather than identity labels or free-form attribution. This design choice intentionally steers the supervision signal toward temporal and causal evidence (what happens, when, and why) and away from sensitive attributes (who a person is), thereby reducing privacy and profiling risks at the data layer. Concretely, we derive questions from captions and temporally grounded observations instead of collecting personal metadata; we avoid tasks that require demographic inference, face recognition, or persistent identity tracking; and we release structured Q&A rather than user-originating personal content. In this way, the dataset construction supports the intended societal benefits of long-video reasoning while mitigating risks such as privacy leakage, surveillance-like use, and stereotype amplification.

In conclusion, LongVILA-R1 demonstrates the potential of long video reasoning to drive progress across a wide range of applications, from robotics to immersive virtual environments. However, unlocking the full promise of this technology requires a steadfast commitment to ethical principles, privacy protection, and the broader goal of benefiting humanity. By addressing these challenges, the AI community can ensure that advancements in long video reasoning contribute positively to society while mitigating associated risks.

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

# Appendix

In this section, we selected examples of four types of reasoning tasks from the benchmark to evaluate and compare the reasoning processes and answers provided by Gemini-1.5-Pro, Video-R1-7B, and LongVILA-R1-7B. On Figure 12, a 20-minute StarCraft match is depicted, where the models analyze the players' unit compositions, strategies, and play styles to predict the potential developments on the battlefield. While Gemini-1.5-Pro produced a correct prediction of the outcome, its reasoning process contained factual inaccuracies. In contrast, Video-R1-7B, influenced by the characteristics of its training data, tended to summarize answers based on options, neglecting critical video details and resulting in incorrect reasoning. LongVILA-R1-7B, however, is able to accurately analyze the players' operational styles and specific moments marked in the video, leading to a comprehensive and accurate prediction of the match's trajectory. On Figure 13, another example demonstrates the models' abilities in narrative reasoning and visual information analysis. Gemini-1.5-Pro failed to correctly infer why the man appearing for the second time in the video is not the husband. In contrast, both Video-R1-7B and LongVILA-R1-7B successfully reasoned that the man's habit of wearing a ring on his left hand is a key indicator, providing accurate answers.

Figure 14 illustrates the models' spatial perception and reasoning abilities as the camera moves through a room. Gemini-1.5-Pro effectively identified the key information within the video and provided the correct answer through straightforward reasoning. In contrast, Video-R1-7B experienced significant localization errors during the reasoning process, leading to a critical issue for reasoning models: a mismatch between the reasoning analysis and the final answer. LongVILA-R1-7B demonstrated superior performance by leveraging dense frame analysis to accurately infer the spatial relationships between rooms and furniture across different levels, ultimately delivering a coherent reasoning process and the correct answer. On Figure 15, the focus shifts to temporal analysis in a Lego video featuring diverse events and interactions. All three models successfully reasoned through the sequence of events and provided correct answers, showcasing their proficiency in temporal reasoning tasks. As a supplement to Figure 1, Figure 10 provides a more comprehensive comparison of two examples: "2022 FIFA Argentina vs. Netherlands" and "Moving the Cup and Guessing Where the Ball Is." In the football match example, while Gemini-1.5-Pro produced the correct answer, its output contained hallucinatory content influenced by biases in its pre-learned knowledge. Video-R1 not only failed to provide accurate video analysis reasoning but also made incorrect predictions. In contrast, LongVILA-R1 successfully analyzed the players' performance and emotions during the match, integrating these factors through its robust reasoning capabilities to make accurate predictions about the outcome. For the more challenging task of tracking the ball, Gemini-1.5-Pro's reasoning is inconsistent with the spatial content throughout, while Video-R1 failed to deduce the ball's final position accurately. Remarkably, LongVILA-R1 precisely analyzed the spatial transformations following the movement of the box, demonstrating superior interpretative and reasoning abilities.

**Temporal 2022 FIFA Argentina v.s. Netherlands**

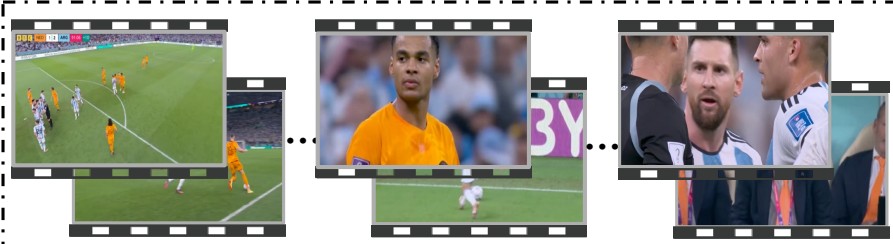

*Question:* You are shown a 30-minute video segment from a football game. The score remains 2–2 throughout this period, and no goals are scored. Based on players' physical condition…which team is more likely to win the upcoming penalty shootout?

    *A: The Netherlands win the 4–2…regulation time.*
    *B: Argentina win the 4–3…maintaining composure.*
    *C: The Netherlands win the shootout 5–4…by Argentina.*
    *D: Argentina score …3–2 without going to penalties*

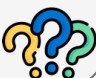

*Gemini-1.5-Pro:* The video shows the extra time of the Netherlands vs Argentina match. Both teams score no goal, leading to a penalty shootout….Based on the video, at 00:11, the narrator points out that the two teams played a draw in the previous tournament eight years ago and went into a penalty shootout, in which the Argentine goalkeeper emerged as the..This suggests that the players were exhausted and playing aggressively. It is pointed out that Argentina has played 9 periods of extra time in the world cup history.
<answer>B</answer>

*Video-R1:* <think>Let's break down the situation step by step:1.Scoreline: The score is tied at 2-2, meaning neither team has won the match yet. This….3. Players' Physical Condition:- Both teams appear to be physically drained after 120 minutes of play. However, the Netherlands might have a slight edge as they were the first to score in the match, potentially giving them a bit more stamina…. They have shown more consistency and have a better chance of converting their penalties.</think>
<answer>A</answer>

*LongVILA-R1:* <think> The score remains 2–2 throughout extra time, indicating a high-intensity match with both teams committed to winning… Argentina's goalkeeper, who initially struggled, regains confidence and makes a crucial save, demonstrating his improved focus and determination. Key players from both teams, such as Messi and de Ligt, are seen holding their heads in disappointment or celebration, reflecting the emotional highs and lows that often decide penalty shootouts… Given the evidence, Argentina is more likely to win the penalty shootout 4–3, with their goalkeeper playing a decisive role and key players maintaining composure. </think>
<answer>B</answer>

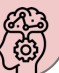

Figure 10: Detailed comparisons in the football game example. The video is available at Link.

## Spatial Track - Where the ball is

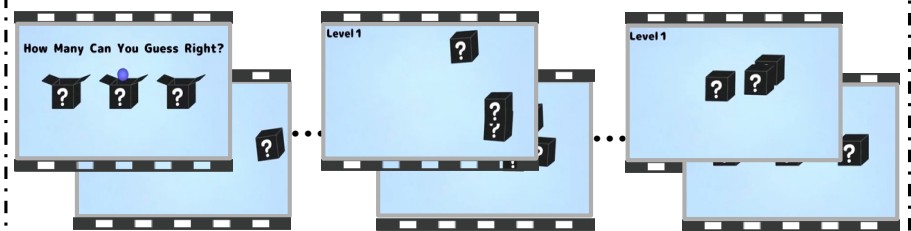

*Question:* There are three boxes, and a purple ball is initially placed in the middle box. Then the positions of the boxes are swapped. Please analyze step by step the movement trajectory of the box containing the ball through the video and determine the final position of the ball.

> *A: Left*  *B: Middle*
> *C: Right*  *D: None of the above*

*Gemini-1.5-Pro:* Here is the step-by-step analysis of the movement of the boxes:1. At the beginning of the video, the purple ball is located in the middle box.2. The middle box moves to the right.3. The right box moves to the middle.4. The middle box moves to the right, then to the left.5. The left box moves to the middle.6. The middle box moves to the left.7. The right box moves to the middle, then to the left. 8. The middle box moves to the right, then to the left. Based on the box movements, we can track that the ball is in the left box.
<answer>A</answer>

*Video-R1:* <think>Let's break down the problem step by step…Now, let's consider the movement of the box containing the ball:- The box that originally had the ball (middle box) will now be in the right position after the swap….Therefore, the ball will be in the right box…original right box moves to the left. The ball is in the box that was originally in the middle, which is now in the right position. Final position of the ball: right box.</think>
<answer>C</answer>

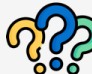

*LongVILA-R1:* <think The middle box initially contains the purple ball. When the boxes are swapped, the left box becomes the new middle box, the middle box moves to the right, and the right box moves to the left. Since the left box is now in the middle position, the ball, which was previously in the middle box, is now in the left box. The swapping continues with the boxes rotating left, but the ball's position does not change because it is always in the left box. Therefore, the final position of the ball is in the left box. </think>
<answer>A</answer>

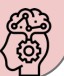

Figure 11: Detailed comparisons in the moving cup example. The video is available at Link.

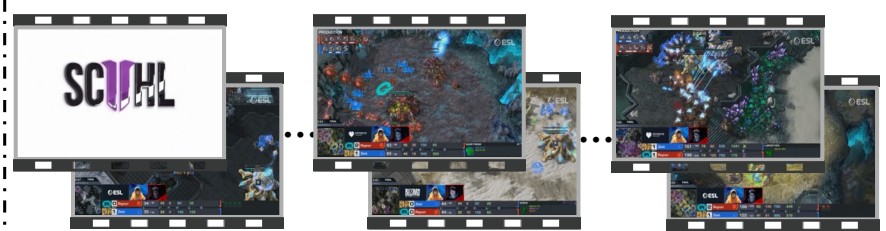

**Goal and Purpose - Strategic Intent Prediction in StarCraft 2**

*Question:* You are presented with the first 20 minutes of a StarCraft II game. What are the likely strategic intentions of both players, their next possible actions, and who is more likely to win the match?
*A: Zest executes a successful Stargate-based air harassment strategy…*
*B: Reynor adapts to Zest's early aggression, transitions into a Roach-Ravager…*
*C: Zest's early proxy strategy catches Reynor off-guard, leading to a quick win…*
*D: Both players engage in a prolonged macro game, but Zest's superior…*

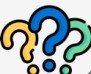

*Gemini-1.5-Pro:* At 0:04, we can see Reynor uses an overlord to scout Zest's base. Reynor starts building Queens in response to Zest showing a void ray. At 0:30, Zest sends an oracle into Reynor's base. Zest's zealots clear out Reynor's early expansion. At 2:21, Zest's void ray damages Reynor's base. At 4:05, Reynor's roach-ravager is quite big. At 6:48, Reynor's ground armor is completed. At 6:57, Reynor defeats Zest….So, options C and D can be eliminated. Reynor adapts to Zest's early aggression and secures victory by securing his economy. Option A is false.
<answer>B</answer>

*Video-R1:* <think>Let's break down the situation step by step: 1. Initial Observations:- The game appears to be at an early stage, with both players having established some form of…Option A suggests Zest's success with the Stargate, which seems plausible given his early advantage…Option C suggests a quick win for Zest, which is less likely due to the early stage and Reynor's potential adaptability.
  - Option D suggests a prolonged game…</think>
<answer>A</answer>

*LongVILA-R1:* <think> The initial frames show Zest using a Stargate-based air harassment strategy, focusing on early aggression with air units like Stalkers…However, by the 15-minute mark, Reynor's economy is stabilizing, and they are building a mix of ground and air units, including Marauders, Overlords, and Zealots. This suggests a transition to a balanced strategy…The final frames highlight Reynor's dominance in the late game, supported by their balanced unit composition, including Roaches, Zealots, and Marauders, which allowed them to adapt to Zest's early strategy and counterattack effectively. Therefore, the correct answer is B: Reynor adapts to Zest's early aggression, transitions into a Roach-Ravager composition, and secures victory through sustained ground assaults. </think>
<answer>B</answer>

Figure 12: LongVILA-R1 reasoning example in the "Goal and Purpose" category. The video is available at Link.

## Plot and Narrative - Detector Game

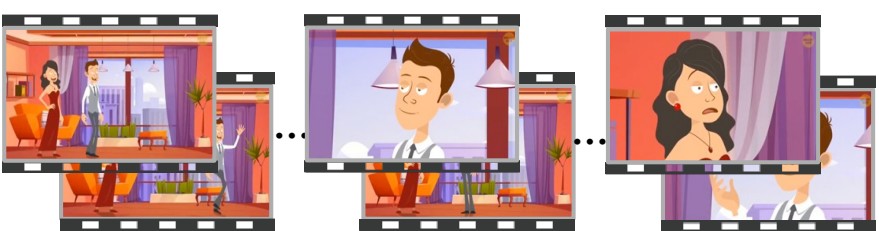

**_Question:_** Dylan and Susan had been happily married for 10 years. One day, Dylan went on a business trip. When he returned, Susan immediately understood that the man was not her husband. How did she figure it out?

*A: This man doesn't look like Dylan*
*B: The man's sleeve is open*
*C: The man is not wearing a wedding ring on his left hand*
*D: Susan is hallucinating*

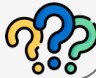

**_Gemini-1.5-Pro:_** Here are the steps to think about this riddle:\n\n1. The video shows a man greeting Susan. This man seems to look like Dylan at first glance. Option A can be eliminated…4. Then Susan keeps looking at the man's left hand. She is supposed to check if the ring is on the right finger. Option C can be eliminated. \n\n5. The man opens another sleeve at 0:22. He wears his wedding ring on his right hand. He is not Dylan. The answer is B.

**_Video-R1:_** <think>Let's break down the situation step by step: 1. Dylan and Susan have been married for 10 years, which suggests a strong bond and familiarity…B. "The man's sleeve is open." - This detail is not specific enough to determine if the man is the husband. It could be a coincidence or a result of casual behavior…context and the details provided, the most logical and specific indicator that Susan figured out the man was not her husband is the absence of a wedding ring on his left hand. </think>
<answer>C</answer>

**_LongVILA-R1:_** <think> Susan recognized the man as her husband because Dylan wears a wedding ring, which symbolizes marital status. In the video, the man is seen wearing a ring on his left hand while standing in front of a window. This detail indicates his marital status…he is also seen wearing a gray suit and a white shirt, but there is no mention of a wedding ring…The key detail is that Dylan wears a wedding ring, which Susan would recognize as her husband's symbol of marriage. The man's appearance, attire, and lack of a wedding ring are distractions. The correct answer is based on the symbolic importance of the ring, which confirms Dylan's identity as her husband. </think>
<answer>C</answer>

Figure 13: LongVILA-R1 reasoning example in the "Plot and Narrative" category. The video is available at Link.

## Spatial – House Tour

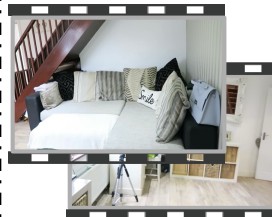 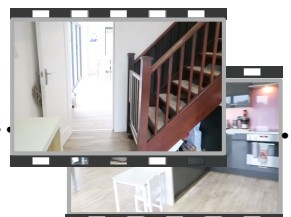 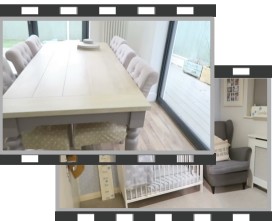

***Question:*** Based on the video, which room is most likely located on the upper floor adjacent to the staircase with the wedding photo, requiring an understanding of spatial and temporal progression across multiple scenes?
***A:*** *Kitchen with gray cabinets and pink backsplash*
***B:*** *Nursery with a white crib and light blue walls*
***C:*** *Bathroom with a white bathtub and floral wallpaper*
***D:*** *Dining room with a rectangular wooden table*

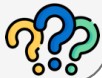

***Gemini-1.5-Pro:*** Here are the thinking steps:
1. Locate the staircase with the wedding photo. The staircase appears on the screen at [00:00:48]. The wedding photo is shown at [00:00:50].
2. Identify the rooms adjacent to the staircase with the wedding photo. The adjacent room appears at [00:00:52]. It contains a crib and blue wallpaper with small white stars. Emily refers to this room as the "baby's room".
<answer>B</answer>

***Video-R1:*** <think>Let me think about this carefully…This requires us to consider the spatial layout and the sequence of rooms shown in the video…This room is also shown in the video, but it's not clear if it's on the upper floor or not…This room is shown in the video, but it's not clear if it's on the upper floor or not. Oh, I see. The kitchen is the most logical choice based on the spatial layout and the fact that it's a common room on the upper floor…Final answer: A Let's break it down one last time to ensure. </think>
<answer>B</answer>

***LongVILA-R1:*** <think> The staircase has a black railing on the left and a white wall on the right, with a wedding photo visible on the right wall…adjacent to the living room. The bathroom with a white bathtub and floral wallpaper is described as having a window with a white curtain, and it is never mentioned to be on the first floor. The nursery with a white crib and light blue walls is described as having a staircase with a black railing on the left, matching the staircase in the first scene, and it is explicitly stated to be on the second floor. Therefore, the bathroom and kitchen are on the first floor, while the nursery is on the second floor adjacent to the staircase with the wedding photo. </think>
<answer>B</answer>

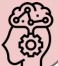

Figure 14: LongVILA-R1 reasoning example in the "Spatial" category. The video is available at Link.

## Temporal – LEGO PINATA

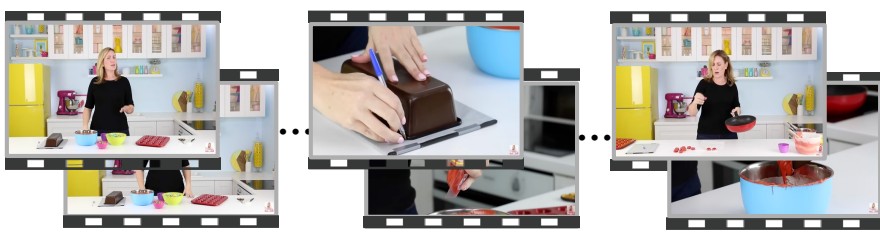

**Question:** Which cake's preparation process involved both the use of a blue hammer for smashing and a red silicone mold for shaping decorative elements, based on the temporal progression of tools and ingredients observed in the video?
*A: Green/yellow/red Lego block cakes*
*B: Red velvet cake with batter mixed in a blue bowl*
*C: Chocolate cake with parchment paper and red cake pops*
*D: Lego-themed cake smashed at 0:00:30-0:00:40 and decorated with bricks*

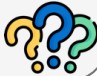

**Gemini-1.5-Pro:** The question is about the cake's preparation process shown in the video…The yellow Lego-themed cake is smashed with an orange hammer. The green Lego-themed cake is opened with a knife. 3. At 0:30-0:40, red silicone mold is used for shaping a decorative brick by pouring melted chocolate into its cases. 4. So, both tools are present only at the preparation of the red Lego-themed cake in the video.
<answer>D</answer>

**Video-R1:** <think>Let me think about this carefully…First, let's consider the options:
A: Green/yellow/red Lego block cakes - This option mentions Lego blocks, but there's no mention of a blue hammer or a red silicon…after going through all the options, it's clear that the correct choice is D…. Let's verify one last time to make sure. Yes, the video indeed shows the use of a blue hammer and a red silicone mold in the preparation of the Lego-themed cake. So, the final answer is D</think>
<answer>D</answer>

**LongVILA-R1:** <think> The preparation of the Lego-themed cake involved the use of a blue hammer to smash colorful Lego blocks, which were then used as decorations. The red silicone mold was used to shape the red decorative elements for the cake. The process started with the woman using a blue hammer to smash green, yellow, and red Lego blocks, which confirmed the use of the blue hammer. The red silicone mold was explicitly mentioned when shaping the red decorative elements. The cake was then decorated with bricks, indicating the use of the red silicone mold for the red elements…The red velvet cake used a blue bowl for mixing batter but did not use the blue hammer or red silicone mold. Therefore, the correct answer is D.</think>
<answer>D</answer>

Figure 15: LongVILA-R1 reasoning example in the "Temporal" category. The video is available at Link.

# Taboo Game

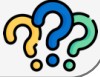

**Question:** This a video of two people playing the game Taboo on the topic of 'machine learning.' One person describes a word, and the other person guesses it. Your task is to analyze the video and choose the most likely word being described from the following options:

**A:** *Logistic Regression - Pruning - Attention Mechanism*
**B:** *Support Vector Machine - Quantization - Transformer*
**C:** *Neural Network - Pruning - Gradient Boosting*
**D:** *Neural Network - Quantization - Transformer*

**Gemini-1.5-Pro:** Here are the thinking steps:1. First Clue: The description mentions similarity to the human brain, layers of nodes, weighted connections, and origin in the 1950s. This strongly suggests "Neural Network." All other options in the sets don't have these defining features…3. Third Clue: The clue references a science fiction film with characters from Cybertron, Optimus Prime, and Megatron. This leads to "Transformer," a powerful neural network architecture, and its naming convention evokes the Transformers franchise.
<answer>D</answer>

**Video-R1:** <think>The question involves analyzing the context provided by the video and matching it with the given options. The participants are discussing machine learning concepts, which narrows down the focus. Firstly, let's consider the options… Neural Network - Quantization – Transformer... After analyzing the options, B seems the most balanced and relevant to the context of the video. The participants are likely discussing a mix of fundamental and advanced concepts in machine learning.\.</think>
<answer>B</answer>

**LongVILA-R1:** <think> The word "Logistic Regression" is not actually mentioned; instead, the description "similar to the human brain, with a lot of layers of nodes and weighted connections" clearly refers to a Neural Network, a fundamental machine learning model originating in the 1950s. The second clue, about "cutting the digits of an algorithm and localizing it to an edge device," matches Quantization, which reduces bit-width to make models more efficient. The mention of "Optimus Prime and Megatron" directly points to Transformer, named after the Transformers franchise and used widely in modern NLP. Together, Neural Network, Quantization, and Transformer form a coherent chain of concepts matching the video's clues. Options A and C omit Transformer or include unrelated terms. Therefore, the best answer is D.</think>
<answer>D</answer>

Figure 16: LongVILA-R1 reasoning example in the taboo game. The video is available at Link.

