# OpenReview forum: "Scaling RL to Long Videos"
_NeurIPS.cc/2025/Conference — NeurIPS 2025 poster_

### Official Review · Reviewer_XSh3 · 2025-06-24

**Clarity:** 2
**Significance:** 3
**Originality:** 2
**Rating:** 4
**Confidence:** 3

**Summary:**

The paper describes a dataset, benchmark and RL-based training methodology for reasoning over long videos. The dataset is created by automatically generating annotations (using existing foundation models) to videos from an existing video dataset. A model trained using the introduced data and training methodology is evaluated on the VideoMME benchmark as well as on the newly introduced benchmark. It shows that the resulting model outperforms several existing models.

**Questions:**

- The results of LongVILA [3] are missing from Table 2.
- Why is the 64-frames result from Video-R1-7B [12] not shown or mentioned?
- Will the code for data generation and the MR-SP training strategy be made available?
- "question-answer pairs are generated via a leading open-source reasoning LLM" - which one?
- Line 52: "We also manually curate a balanced set of 1K long-video samples" - what does "balanced" here mean?
- (small comment: the soccer example in Figure 1 is strange in that the prediction to be made is very stochastic and model correctness will be almost random)

**Ethical Concerns:**

["NO or VERY MINOR ethics concerns only"]

**Final Justification:**

Authors made significant improvements during the rebuttal period, addressing my concerns, and resulting in a much stronger paper than the original submission. In the current form, the method and dataset should be useful contribution to the emerging area of reasoning in long videos.

**Limitations:**

Yes

**Quality:**

3

**Strengths And Weaknesses:**

Strengths:
- The paper addresses a difficult (and increasingly popular) task: reasoning in long videos.
- It addresses the task by introducing a new (automatically generated) dataset and a multi-GPU training strategy supporting RL training on long videos.
- The paper also introduces a new hand-curated benchmark, which can be useful for revaluating future models (assuming it will be made publicly available, see below).

Weaknesses:
- The performance of the newly introduced model is very similar to (and only marginally higher than) LongVILA [3], of which this is in a sense a derivative work. For example, results on MME reported in [3] are 60.1 (vs 60.3 in this work) in the "w/o subtitle"-setting and 65.1 (vs 65.9) in the "w subtitle"-setting.
- The performance of the model is actually lower than that of Video-R1-7B reported in [12], which is 61.4 in the 64-frames-setting (Table 1 in [12]). Instead, only the performance of the lower performing 16-frames setting of that work is shown (which is 59.3). Why is that? Am I missing something here?
- By comparison, the only strong performance over existing methods is on the own (1000 example-)benchmark.
- The paper is fairly hard to read due to typos, inconsistencies, and overly touting language. (To name a few examples: Line 44: "long-video understandings", Line 44: "we strategically construct a high-quality dataset", Figure 3 caption: "..., from which 52K QAs with reasoning annotations in LongVideo-Reason.", Line 111 "We then apply a high-quality automated CoT annotation pipeline", ...

---

> ### Author Rebuttal · Authors · 2025-07-31
>
> ## **General Response**
>
> We sincerely appreciate your detailed and constructive feedback. Before addressing each concern individually, we first to highlight the major updates as follows.
>
> 1. **Larger Dataset:** We double the size of our dataset, LongVideo-Reason, from 52K to 104K reasoning data, with carefully filtering and data cleaning. Moreover, we include both multi-choice and open-ended QAs, making it more general.
> 2. **VideoMME Update**: Thanks to the updated dataset, our model achieved strong performance, e.g., **65.1% / 71.1%** on VideoMME [1] without and with subtitles, outperforming the LongVILA [2] baseline, which achieved 60.1% / 65.1% on VideoMME [1].
> 3. **Expanded Evaluation**: We have included **7 additional video benchmarks** for evaluation: ActivityNet-QA [3], PerceptionTest [4], VNBench [5],  LongVideoBench [6], EgoSchema [7],  NExT-QA [8], and MVBench [9]. Alongside VideoMME [1], we now compare LongVILA-R1 against LongVILA [2] baseline, and other methods [10, 11, 12, 13 ,14, 15] on a total of **8 benchmarks**, demonstrating consistently strong performance. Results are show in the table below.
> 4. **Extended Inputs**: LongVILA-R1-7B is capable of processing **2,048** video frames per video, with customizable FPS settings.
>
> | Model | VideoMME (w/o sub) | VideoMME (w/ sub) | ActivityNet-QA | PerceptionTest | VNBench | LongVideoBench | EgoSchema | NExT-QA | MVBench |
> | --- | --- | --- | --- | --- | --- | --- | --- | --- | --- |
> | VIdeo-LLaVA-7B | 39.9 | 41.6 | 45.3 | - | 12.4 | 37.6 | 38.4 | - | 43.5 |
> | ShareGPT4Video-8B | 39.9 | 43.6 | 50.8 | - | - | 41.8 | - | - | 51.2 |
> | VideoLLaMA2.1-7B | 54.9 | 56.4 | 53.0 | 54.9 | - | - | 53.1 | - | 57.3 |
> | Kangaroo-8B | 56.0 | 57.6 | - | - | - | 54.8 | 62.7 | - | 61.1 |
> | LLaVA-OV-7B | 58.2 | 61.5 | 56.7 | 57.1 | 51.8 | 56.4 | 60.1 | 79.4 | 56.7 |
> | Video-R1-7B | 61.4 | - | - | - | - | - | - | - | 64.8 |
> | LongVILA-7B | 60.1 | 65.1 | 59.5 | 58.1 | 63.0 | 57.1 | 67.7 | 80.7 | 67.1 |
> | LongVILA-R1-7B | **65.1** | **71.1** | **64.8** | **68.9** | **75.5** | **58.0** | **68.9** | **81.5** | **68.2** |
>
> ## **Detailed Response for Each Question**
>
> We provide detailed responses to each of the concerns below.
>
> **Q1: “The performance is similar compared to LongVILA”**
>
> **A:** As shown in the table in **General Response**, LongVILA-R1-7B benefits from the updated large-scale dataset, achieving **65.1% / 71.1% on VideoMME** [1] without and with subtitles, representing **+5.0% and +6.0% improvements over LongVILA-7B**, respectively.
>
> **Q2: “Missing Video-R1-7B in the 64-frames-setting”**
>
> **A:** Apologies for the oversight. The 64-frame result of Video-R1-7B [15] (**61.4%** on VideoMME without subtitles) was updated in its v2 version on arXiv on May 14, shortly before the NeurIPS submission deadline, and we missed it. LongVILA-R1-7B achieves **65.1%** on VideoMME without subtitles, surpassing Video-R1-7B. **We have included the comparison in the table in the General Response and will correct this value in the revision.**
>
> **Q3: “The only strong performance on the own benchmark”**
>
> **A:** Thank you for the reminder. As shown in the table in **General Response**, we have evaluated LongVILA-R1 on seven additional benchmarks. Alongside VideoMME [1], it demonstrates strong performance across eight benchmarks in total, for example, **64.8% on ActivityNet-QA** [3], **68.9% on PerceptionTest** [4], and **75.5% on VNBench** [5], showing significant improvements over existing models.
>
> **Q4: “Typos, inconsistencies, and overly touting”**
>
> **A:** Thank you for pointing this out. We will correct the typos and carefully refine the language in our revision to address inconsistencies and overly promotional phrasing.
>
> **Q5: “The results of LongVILA are missing from Table 2”**
>
> **A:** Thank you for the reminder. The results of the LongVILA [2] baseline are provided in the table below, and we will include them in the revision.
>
> | Model | Temporal | Goal | Plot | Spatial | Overall |
> | --- | --- | --- | --- | --- | --- |
> | LongVILA-7B | 58.0 | 80.2 |  57.1 | 46.7 | 62.7 |
> | LongVILA-R1-7B | 68.1 | 85.7 | 70.6 | 53.3 | 72.0 |
>
> **Q6: “Will the code for data generation and the MR-SP training strategy be made available?”**
>
> **A:** Yes, we are fully committed to release the training data, benchmark, data generation pipeline, and the whole training pipeline with the MR-SP training system.
>
> **Q7: “The leading open-source reasoning LLM”**
>
> **A:** It is the DeepSeek-R1-671B [16] model, and we will clarify this in our revision.
>
> **Q8: “What does ‘balanced’ here mean?”**
>
> **A:** By "balanced," we mean that the number of questions is balanced across the four categories: temporal, goal and purpose, plot and narrative, and spatial.
>
> **Q9: “Small comment on the soccer example“**
>
> **A:** Thank you for the suggestion. We will include additional examples in the revised version to better demonstrate the advantages of our model.
>
> ### **References**
>
> [1] Video-MME: The First-Ever Comprehensive Evaluation Benchmark of Multi-modal LLMs in Video Analysis. CVPR 2025
>
> [2] LongVILA: Scaling Long-Context Visual Language Models for Long Videos. ICLR 2025
>
> [3] ActivityNet-QA: A Dataset for Understanding Complex Web Videos via Question Answering. AAAI 2019
>
> [4] Perception Test: A Diagnostic Benchmark for Multimodal Video Models. NeurIPS 2023
>
> [5] Needle In A Video Haystack: A Scalable Synthetic Framework for Benchmarking Video MLLMs. ICLR 2025
>
> [6] LongVideoBench: A Benchmark for Long-context Interleaved Video-Language Understanding. NeurIPS 2024
>
> [7] EgoSchema: A Diagnostic Benchmark for Very Long-form Video Language Understanding. NeurIPS 2023
>
> [8] NExT-QA: Next Phase of Question-Answering to Explaining Temporal Actions. CVPR 2021
>
> [9] MVBench: A Comprehensive Multi-modal Video Understanding Benchmark. CVPR 2024
>
> [10] Video-LLaVA: Learning United Visual Representation by Alignment Before Projection. EMNLP 2024
>
> [11] ShareGPT4Video: Improving Video Understanding and Generation with Better Captions. NeurIPS 2024
>
> [12] VideoLLaMA 2: Advancing Spatial-Temporal Modeling and Audio Understanding in Video-LLMs. ArXiv 2024
>
> [13] Kangaroo: A Powerful Video-Language Model Supporting Long-context Video Input. ArXiv 2024
>
> [14] LLaVA-OneVision: Easy Visual Task Transfer. Trans. Mach. Learn. Res. 2025
>
> [15] Video-R1: Reinforcing Video Reasoning in MLLMs. ArXiv 2025
>
> [16] DeepSeek-R1: Incentivizing Reasoning Capability in LLMs via Reinforcement Learning. ACL 2025

---

> > ### Comment · Reviewer_XSh3 · 2025-08-05
> >
> > I thank the authors for their comprehensive response, which addresses my concerns. Quantitative results have also improved significantly as a result of the updates.
> >
> > "we are fully committed to release the training data, benchmark, data generation pipeline, and the whole training pipeline with the MR-SP training system."
> > I read this as a "yes" (as in "will be released").
> >
> > The comprehensive updates will significantly improve the paper, and I will accordingly raise my rating.

---

> > > ### Author Response · Authors · 2025-08-05
> > >
> > > Dear Reviewer XSh3,
> > >
> > > Thank you for your kind response.
> > >
> > > We **confirm that we will release all relevant materials**, including the training data, benchmark, data generation pipeline, and the full training framework with the MR-SP system.
> > >
> > > We sincerely appreciate your recognition of our work and your willingness to consider raising the rating.
> > >
> > > Sincerely,
> > >
> > > Authors of Paper 8328

---

> ### Author Response · Authors · 2025-08-05
> **Follow-up on Rebuttal for Paper 8328**
>
> Dear Reviewer XSh3,
>
> Thank you once again for your constructive and insightful review, which has significantly helped improve the quality of our work.
>
> Following your valuable suggestions, we have revised our paper and submitted a detailed rebuttal. In summary, we have:
>
> 1. Conducted **comprehensive comparisons**, including evaluations against **LongVILA** across 8 benchmarks with non-trivial improvements, and comparisons with **Video-R1 (64-frame setting)** on VideoMME (w/o sub) (see **General Response, Q1, Q2, Q3, Q5**).
> 2. Provided **additional clarifications** on typos, code release, and other details (see **Q4, Q6, Q7, Q8, Q9**).
>
> We hope these updates have addressed your concerns. As the discussion period is drawing to a close, we would greatly appreciate it if you could let us know whether our response has resolved your questions.
>
> Thank you again for your time and consideration.
>
> Sincerely,
>
> Authors of Paper 8328

---

### Official Review · Reviewer_oP4r · 2025-07-03

**Clarity:** 2
**Significance:** 2
**Originality:** 2
**Rating:** 4
**Confidence:** 3

**Summary:**

This paper presents LongVILA-R1: a framework for long video reasoning in VLMs and introduces a new dataset (LongVideo-Reason). The proposed LongVILA-R1 framework consists of a two-stage CoT-SFT and RL training approach. The paper also introduces MR-SP, for efficient for scalable long video RL. The evaluation is performed on the proposed LongVideo-Reason dataset and Video-MME.

**Questions:**

•	The paper should clarify the novelty of the training scheme in more detail.

•	The paper should provide more detailed analysis of the diversity of the proposed LongVideo-Reason dataset.

•	The paper should clarify the choice of evaluation datasets and baseline methods.

**Ethical Concerns:**

["NO or VERY MINOR ethics concerns only"]

**Final Justification:**

The rebuttal addressed by concerns.

**Limitations:**

Yes, but the limitations of the evaluation scheme should also be discussed.

**Paper Formatting Concerns:**

None.

**Quality:**

2

**Strengths And Weaknesses:**

**Strengths:**

•	The LongVideo-Reason dataset is diverse and contains complex video reasoning questions, e.g., tracking balls in a shell game (Figure 12).

•	The paper introduces a novel and interesting training paradigm (Multi-modal Reinforcement Sequence Parallelism) that shows promising improvements in training complexity.

•	The paper shows promising performance on the proposed LongVideo-Reason-eval benchmark and other datasets such as Video-MME.

**Weaknesses:**

•	The proposed LongVideo-Reason dataset is based only on the Shot2Story dataset. As the video score is a single dataset, this might limit the diversity of the LongVideo-Reason dataset. The paper should discuss the diversity of the LongVideo-Reason dataset in more detail.

•	Novelty: The dataset generation and training scheme – supervised fine-tuning followed by GRPO based RL training is largely based on prior work, e.g., the Deepseek family of models. The novelty of the training scheme should be discussed in more detail.

•	Evaluation on Video-MME: The paper does not compare the state of the art approaches such as VideoLLaMA3 which achieves better results -- 54.9% accuracy on long videos without subtitles – without the need of RL based training.

•	Evaluation on LongVideo-Reason-eval: While the proposed model outperforms the Gemini-1.5-Pro and GPT-4o models on this benchmark, the proposed model was already fine-tuned on this dataset. This raises the question whether the comparison with zero-shot methods such as Gemini-1.5-Pro and GPT-4o is fair.

•	Long videos: The proposed method seems to be limited to 512 frames, while state of the art approaches, e.g., Qwen-2.5-VL can process >=1200 frames.

•	The model seems to rely on a stable and low frame rate, it is unclear if the model can deal with variable or high frame rates.

---

> ### Author Rebuttal · Authors · 2025-07-31
>
> ## **General Response**
>
> We sincerely appreciate your detailed and constructive feedback. Before addressing each concern individually, we first to highlight the major updates as follows.
>
> 1. **Larger Dataset:** We double the size of our dataset, LongVideo-Reason, from 52K to 104K reasoning data, with carefully filtering and data cleaning. Moreover, we include both multi-choice and open-ended QAs, making it more general.
> 2. **VideoMME Update**: Thanks to the updated dataset, our model achieved strong performance, e.g., **65.1% / 71.1%** on VideoMME [1] without and with subtitles, outperforming the LongVILA [2] baseline, which achieved 60.1% / 65.1% on VideoMME [1].
> 3. **Expanded Evaluation**: We have included **7 additional video benchmarks** for evaluation: ActivityNet-QA [3], PerceptionTest [4], VNBench [5],  LongVideoBench [6], EgoSchema [7],  NExT-QA [8], and MVBench [9]. Alongside VideoMME [1], we now compare LongVILA-R1 against LongVILA [2] baseline, and other methods [10, 11, 12, 13 ,14, 15] on a total of **8 benchmarks**, demonstrating consistently strong performance. Results are show in the table below.
> 4. **Extended Inputs**: LongVILA-R1-7B is capable of processing **2,048** video frames per video, with customizable FPS settings.
>
> | Model|VideoMME (w/o sub)|VideoMME (w/ sub)|ActivityNet-QA|PerceptionTest|VNBench|LongVideoBench|EgoSchema|NExT-QA|MVBench|
> |---|---|---|---|---|---|---|---|---|---|
> |VIdeo-LLaVA-7B|39.9|41.6|45.3|-|12.4|37.6|38.4|-|43.5|
> |ShareGPT4Video-8B|39.9|43.6|50.8|-|-|41.8|-|-|51.2|
> |VideoLLaMA2.1-7B|54.9|56.4|53.0|54.9|-|-|53.1|-|57.3|
> |Kangaroo-8B|56.0|57.6|-|-|-|54.8|62.7|-|61.1|
> |LLaVA-OV-7B|58.2|61.5|56.7|57.1|51.8|56.4|60.1|79.4|56.7|
> |Video-R1-7B|61.4|-|-|-|-|-|-|-|64.8|
> |LongVILA-7B|60.1|65.1|59.5|58.1|63.0|57.1|67.7|80.7|67.1|
> |LongVILA-R1-7B|**65.1**|**71.1**|**64.8**|**68.9**|**75.5**|**58.0**|**68.9**|**81.5**|**68.2**|
>
> ## **Detailed Response for Each Question**
>
> We provide detailed responses to each of the concerns below.
>
> **Q1: “The diversity of the LongVideo-Reason dataset”**
>
> **A:** Apologies for the confusion. The videos in LongVideo-Reason are not solely sourced from the Shot2Story [16] dataset. We also incorporated an additional **2,000 4K-resolution** videos covering diverse scenarios, including **autonomous driving, video games, household robotics, and wildlife**. We used VILA-HD [17] to generate object bounding boxes within video frames and constructed spatial reasoning Q&A pairs based on these boxes and their corresponding captions. We will add detailed description of this in the revision.
>
> The diversity of the LongVideo-Reason dataset also stems from its reasoning-annotated QA pairs, which are categorized into four reasoning types: **Temporal**, **Goal and Purpose**, **Spatial**, and **Plot and Narrative**. After thorough filtering and cleaning, the LongVideo-Reason dataset contains **104K** QA pairs with explicit reasoning annotations. Representative visual examples are provided in Figures 10 and 11.
>
> Additionally, the dataset contains a balanced mix of **multiple-choice** and **open-ended questions**, enhancing its versatility. As a result, as shown in the **General Response**, this dataset significantly contributes to the model’s strong performance across multiple benchmarks, which cover a diversity of video tasks.
>
> **Q2: “The novelty of the dataset generation and the training scheme”**
>
> **A:** We answer this from two perspectives:
>
> **(1) Dataset Generation**
>
> The novelty of our dataset generation lies in the **pipeline design**: we first use VLMs to generate text captions for video frames, and then leverage powerful reasoning-oriented LLMs to generate multi-step reasons. While advanced LLMs like DeepSeek-R1-671B [18] possess strong reasoning capabilities, they cannot process video inputs. Our pipeline bridges the gap between long-video VLMs and the reasoning-oriented LLMs.
>
> Different from existing video datasets [19] which only provide QAs, our dataset includes detailed reasons for QA pairs. While some datasets do include reasoning annotations, they are designed for LLMs, such as mathematics [20], not videos.
>
> Moreover, this dataset helps LongVILA-R1-7B to achieve strong performance (e.g., 65.1/71.1 on VideoMME) and serves as a solid baseline for future research, making it a meaningful contribution to the field.
>
> **(2) Training Scheme**
>
> The novelty of our training scheme lies in **the co-design of algorithm and infrastructure**, even though the SFT+RL algorithm itself is not new. A central component of this scheme is the **MR-SP** system, which is carefully tailored for efficient long-video RL. It features several key designs:
>
> 1. We first implemented **video encoding with balanced distribution** across multiple GPUs. (Traditional RL frameworks often run into **GPU OOM** issues when processing more than 512 frames)
> 2. We introduce a **chunked distributed gathering** strategy for multimodal inputs and embeddings. (Traditional RL frameworks often suffer from **CPU OOM** issues due to the combination of long video frames and a large number of rollouts)
> 3. We further **cache and reuse video embeddings** during both the rollout and pre-filling phases, effectively reducing redundant computation. (Traditional RL frameworks often suffer from **high latency** due to repetitive processing)
>
> Without this co-design, the training algorithm alone is insufficient to handle long-video training effectively. In contrast, our training framework is capable of training on 3,600 video frames using a single node with 8 A100 GPUs.
>
> We are fully committed to releasing the dataset, the complete code of dataset generation pipeline, and the training framework, including the MR-SP system, to support future research in the community.
>
> **Q3: “Evaluation on Video-MME”**
>
> **A:** Thanks for your reminder, we include more VLMs comparison on VideoMME, including NVILA-8B-Video [21] (CVPR 2025), Apollo-7B [22] (CVPR 2025), and VideoLLaMA3-7B [23] (ArXiv 2025).
>
> LongVILA-R1 achieves performance comparable to VideoLLaMA3. Although VideoLLaMA3 does not incorporate RL training, it uses **5.72 million samples** during video fine-tuning, Table 4 in VideoLLaMA3 [23]. In contrast, **we use only 0.21 million samples** in total across both SFT and RL, resulting in higher data efficiency and much lower training cost than VideoLLaMA3.
>
> |Model|VideoMME (w/o sub)|VideoMME (w/ sub)|
> |---|---|---|
> |Apollo-7B (CVPR 2025)|61.1|63.3|
> |NVILA-8B-Video (CVPR 2025)|64.2|70.0|
> |VideoLLaMA3-7B (ArXiv 2025)|66.2|70.3|
> |LongVILA-R1-7B|65.1|71.1|
>
> **Q4: “The comparison with Gemini-1.5-Pro and GPT-4o”**
>
> **A:** To address concerns of zero-shot evaluation, we further compare LongVILA-R1-7B with Gemini-1.5-Pro, GPT-4o, and GPT-4o-mini on ActivityNet-QA, VNBench, and VideoMME. LongVILA-R1-7B outperforms Gemini-1.5-Pro and GPT-4o on ActivityNet-QA and VNBench, and inferior to them on VideoMME, where LongVILA-R1-7B surpasses GPT-4o-mini with only 7B parameters.
>
> |Model|ActivityNet-QA|VNBench|VideoMME (w/o sub)|VideoMME (w/ sub)|
> |---|---|---|---|---|
> |GPT-4o-mini|-|-|64.8|68.9|
> |GPT-4o|61.9|64.4|71.9|77.2|
> |Gemini-1.5-Pro|57.5|66.7|75.0|81.3|
> |LongVILA-R1-7B|64.8|75.5|65.0|70.7|
>
> **Q5: “Long video frames”**
>
> **A:** As mentioned in the **General Response**, LongVILA-R1-7B is capable of processing up to **2,048** video frames per video. We progressively increase the input frames when evaluating on VNBench, and observe that performance improves as the frames increases.
>
> |Frames|128|256|512|1024|2048|
> |---|---|---|---|---|---|
> |Accuracy|60.5|69.5|73.8|74.2|75.5|
>
> **Q6: “Variable or high frame rates”**
>
> **A:** LongVILA-R1-7B supports both variable and high FPS settings. We evaluate it on VideoMME using FPS values of 8, 16, and 32, all of which result in strong performance.
>
> |FPS|8|16|32|
> |---|---|---|---|
> |VideoMME (w/o sub)|65.3|64.8|64.9|
> |VideoMME (w/ sub)|70.5|71.1|70.7|
>
> ### **References**
> [1] Video-MME: The First-Ever Comprehensive Evaluation Benchmark of Multi-modal LLMs in Video Analysis. CVPR 2025
>
> [2] LongVILA: Scaling Long-Context Visual Language Models for Long Videos. ICLR 2025
>
> [3] ActivityNet-QA: A Dataset for Understanding Complex Web Videos via Question Answering. AAAI 2019
>
> [4] Perception Test: A Diagnostic Benchmark for Multimodal Video Models. NeurIPS 2023
>
> [5] Needle In A Video Haystack: A Scalable Synthetic Framework for Benchmarking Video MLLMs. ICLR 2025
>
> [6] LongVideoBench: A Benchmark for Long-context Interleaved Video-Language Understanding. NeurIPS 2024
>
> [7] EgoSchema: A Diagnostic Benchmark for Very Long-form Video Language Understanding. NeurIPS 2023
>
> [8] NExT-QA: Next Phase of Question-Answering to Explaining Temporal Actions. CVPR 2021
>
> [9] MVBench: A Comprehensive Multi-modal Video Understanding Benchmark. CVPR 2024
>
> [10] Video-LLaVA: Learning United Visual Representation by Alignment Before Projection. EMNLP 2024
>
> [11] ShareGPT4Video: Improving Video Understanding and Generation with Better Captions. NeurIPS 2024
>
> [12] VideoLLaMA 2: Advancing Spatial-Temporal Modeling and Audio Understanding in Video-LLMs. ArXiv 2024
>
> [13] Kangaroo: A Powerful Video-Language Model Supporting Long-context Video Input. ArXiv 2024
>
> [14] LLaVA-OneVision: Easy Visual Task Transfer. Trans. Mach. Learn. Res. 2025
>
> [15] Video-R1: Reinforcing Video Reasoning in MLLMs. ArXiv 2025
>
> [16] Shot2Story: A New Benchmark for Comprehensive Understanding of Multi-shot Videos. ICLR 2025
>
> [17] Scaling Vision Pre-Training to 4K Resolution. CVPR 2025
>
> [18] DeepSeek-R1: Incentivizing Reasoning Capability in LLMs via Reinforcement Learning. ACL 2025
>
> [19] Video Instruction Tuning With Synthetic Data. ArXiv 2024
>
> [20] Training Verifiers to Solve Math Word Problems. ArXiv 2021
>
> [21] NVILA: Efficient Frontier Visual Language Models. CVPR 2025
>
> [22] Apollo: An Exploration of Video Understanding in Large Multimodal Models. CVPR 2025
>
> [23] VideoLLaMA 3: Frontier Multimodal Foundation Models for Image and Video Understanding. ArXiv 2025

---

> > ### Comment · Reviewer_oP4r · 2025-08-05
> > **Good paper**
> >
> > The rebuttal addressed by concerns and I will update my score.

---

> > > ### Author Response · Authors · 2025-08-06
> > > **Appreciation for Your Feedback**
> > >
> > > Dear Reviewer oP4r,
> > >
> > > Thank you very much for your kind response. We sincerely appreciate your recognition of our work and your willingness to update the review score — it truly means a great deal to us.
> > >
> > > Sincerely,
> > >
> > > Authors of Paper 8328

---

> ### Author Response · Authors · 2025-08-05
> **Follow-up on Rebuttal for Paper 8328**
>
> Dear Reviewer oP4r,
>
> Thank you once again for your constructive and insightful review, which has significantly helped improve the quality of our work.
>
> Following your valuable suggestions, we have revised our paper and submitted a detailed rebuttal. In summary, we have provided:
>
> 1. Detailed discussions on the **diversity of the dataset** as well as the **novelty of the dataset generation process and training scheme** (see **Q1 & Q2**).
> 2. **Comprehensive comparisons** across 8 benchmarks, including VideoMME, along with new evaluations against Gemini-1.5-Pro and GPT-4o (see **General Response, Q3 & Q4**).
> 3. Additional ablation studies on **long video frames** and **variable / high frame rates**.
>
> We hope these updates have addressed your concerns. As the discussion period is drawing to a close, we would greatly appreciate it if you could let us know whether our response has resolved your questions. If there are any remaining concerns, we will make every effort to address them.
>
> Thank you again for your time and consideration.
>
> Sincerely,
>
> Authors of Paper 8328

---

### Official Review · Reviewer_QtJd · 2025-07-04

**Clarity:** 3
**Significance:** 3
**Originality:** 3
**Rating:** 5
**Confidence:** 3

**Summary:**

The key idea of this paper is applying GRPO-style reinforcement learning to improve long video understanding. The authors identify key bottlenecks that limit this and solve them, the three main contributions being (1) a LongVideo-Reason training set (2) Multi-modal Reinforcement Sequence Parallelism (MR-SP) training strategy that allows extremely long videos to be used with many parallel inferences, which is required for GRPO and finally (3) LongVideo-Reason-eval benchmark, a handpicked benchmark with 1000 samples. The final result is LongVILA-R1-7B LLM, which can ingest up to 512 video frames while still being able to carry out spatio-temporal reasoning.

The LongVideo-Reason training corpus supplies 52 K multiple-choice question–answer pairs (with chain-of-thought) drawn from 18 077 long videos, evenly covering Temporal, Goal & Purpose, Spatial, and Plot & Narrative reasoning. Heuristics are used to split the training data into easy, medium and hard subsets.

LongVILA-R1 is trained in two stages: (i) chain-of-thought supervised fine-tuning on "easy" and "hard" examples, followed by (ii) Group-Relative Policy Optimisation (GRPO) reinforcement learning on "medium" cases . Stage 1 is a SFT-CoT stage designed to "warmup" the model's reasoning capabilities. In Stage 2, the authors also add data from Video-R1, a prior work, to scale up RL.  To keep RL feasible, the paper introduces Multi-modal Reinforcement Sequence Parallelism (MR-SP), which caches video embeddings and shards giant token contexts so policy and reference models prefill in parallel.

For evaluation, the authors curated LongVideo-Reason-eval, 1,000 hand-checked problems that mirror the four reasoning categories in the training dataset. LongVILA-R1 averages 67.9% accuracy, exceeding GPT-4o performance. On prior benchmarks like VideoMME, it outperforms open source VideoLLMs by a large margin.

**Questions:**

Result on standard video benchmarks like MVBench, ApolloBench etc would be appreciated.

As dataset is a big part of the contribution, the authors should give more details and visualizations of it, so we can understand the diversity of the dataset.

**Ethical Concerns:**

["NO or VERY MINOR ethics concerns only"]

**Final Justification:**

The expanded results on video benchmarks made me more confident of the results, the consistent improvement across datasets shows the value of the method better.

**Limitations:**

Yes

**Quality:**

3

**Strengths And Weaknesses:**

Strengths

+ The dataset contributions are valuable to the Video LLM community
+ The overall execution is thorough and polished, resulting in strong performance across benchmarks

Weaknesses

- Evaluations on more Video understanding benchmarks which are common in the VideoLLMs literature would be appreciated
- The primary technical contributions are in the Sequence Parallel training, the architecture and training methods (GRPO) are all just directly applied.

---

> ### Author Rebuttal · Authors · 2025-07-31
>
> ## **General Response**
>
> We sincerely appreciate your detailed and constructive feedback. Before addressing each concern individually, we first to highlight the major updates as follows.
>
> 1. **Larger Dataset:** We double the size of our dataset, LongVideo-Reason, from 52K to 104K reasoning data, with carefully filtering and data cleaning. Moreover, we include both multi-choice and open-ended QAs, making it more general.
> 2. **VideoMME Update**: Thanks to the updated dataset, our model achieved strong performance, e.g., **65.1% / 71.1%** on VideoMME [1] without and with subtitles, outperforming the LongVILA [2] baseline, which achieved 60.1% / 65.1% on VideoMME [1].
> 3. **Expanded Evaluation**: We have included **7 additional video benchmarks** for evaluation: ActivityNet-QA [3], PerceptionTest [4], VNBench [5],  LongVideoBench [6], EgoSchema [7],  NExT-QA [8], and MVBench [9]. Alongside VideoMME [1], we now compare LongVILA-R1 against LongVILA [2] baseline, and other methods [10, 11, 12, 13 ,14, 15] on a total of **8 benchmarks**, demonstrating consistently strong performance. Results are show in the table below.
> 4. **Extended Inputs**: LongVILA-R1-7B is capable of processing **2,048** video frames per video, with customizable FPS settings.
>
> | Model | VideoMME (w/o sub) | VideoMME (w/ sub) | ActivityNet-QA | PerceptionTest | VNBench | LongVideoBench | EgoSchema | NExT-QA | MVBench |
> | --- | --- | --- | --- | --- | --- | --- | --- | --- | --- |
> | VIdeo-LLaVA-7B | 39.9 | 41.6 | 45.3 | - | 12.4 | 37.6 | 38.4 | - | 43.5 |
> | ShareGPT4Video-8B | 39.9 | 43.6 | 50.8 | - | - | 41.8 | - | - | 51.2 |
> | VideoLLaMA2.1-7B | 54.9 | 56.4 | 53.0 | 54.9 | - | - | 53.1 | - | 57.3 |
> | Kangaroo-8B | 56.0 | 57.6 | - | - | - | 54.8 | 62.7 | - | 61.1 |
> | LLaVA-OV-7B | 58.2 | 61.5 | 56.7 | 57.1 | 51.8 | 56.4 | 60.1 | 79.4 | 56.7 |
> | Video-R1-7B | 61.4 | - | - | - | - | - | - | - | 64.8 |
> | LongVILA-7B | 60.1 | 65.1 | 59.5 | 58.1 | 63.0 | 57.1 | 67.7 | 80.7 | 67.1 |
> | LongVILA-R1-7B | **65.1** | **71.1** | **64.8** | **68.9** | **75.5** | **58.0** | **68.9** | **81.5** | **68.2** |
>
> ## **Detailed Response for Each Question**
>
> We provide detailed responses to each of the concerns below.
>
> **Q1: “Evaluations on more benchmarks”**
>
> **A:** Thank you for your suggestion. We fully agree on the importance of evaluating the performance of our reasoning model on a more comprehensive set of benchmarks. As mentioned in **General Response**, we evaluated LongVILA-R1 on eight benchmarks, including VideoMME (w/o sub), VideoMME (w/ sub)  [1], ActivityNet-QA [3], PerceptionTest [4], VNBench [5], LongVideoBench [6], EgoSchema [7], NExT-QA [8], and MVBench [9]. Under these benchmarks, LongVILA-R1 consistently outperforms LongVILA [2] and demonstrates more comprehensive imporvements compared to other video VLMs with similar parameter scales.
>
> **Q2: “The primary technical contributions are in the Sequence Parallel training, while the architecture and training methods (GRPO) are directly applied.”**
>
> **A:** We really appreciate the reviewer's acknowledgement of one of our core contributions, MR-SP system, which enable training on long videos with up to 3,600 frames using a single node with 8 A100 GPUs.
>
> In addition to the technical contributions of MR-SP, our work also provides a dataset, a benchmark, and, more importantly, **a comprehensive pipeline for generating large-scale datasets tailored for long-video RL training**. We believe this contribution has a big value in this area.
>
> The novelty of our training scheme lies in **the co-design of the algorithm and infrastructure**, even though the SFT+RL algorithm itself is not new. The infrastructure features several key design elements:
>
> 1. We first implemented **video encoding with balanced distribution** across multiple GPUs. (Traditional RL frameworks often run into **GPU OOM** issues when processing more than 512 frames.)
> 2. We introduce a **chunked distributed gathering** strategy for multimodal inputs and embeddings. (Traditional RL frameworks often suffer from **CPU OOM** issues due to the combination of long video frames and a large number of rollouts.)
> 3. We further **cache and reuse video embeddings** during both the rollout and pre-filling phases, effectively reducing redundant computation. (Traditional RL frameworks often suffer from **high latency** due to repetitive processing.)
>
> Without this co-design, the training algorithm alone is insufficient to handle long-video training effectively. In contrast, our training framework is capable of training on 3,600 video frames using a single node with 8 A100 GPUs.
>
> We are fully committed to releasing the dataset, the complete code of the dataset generation pipeline, and the training framework, including the MR-SP system, to support future research in the community.
>
> **Q3: “Result on standard video benchmarks like MVBench, ApolloBench”**
>
> **A:** Thank you for the reminder. As mentioned above, we have reported results on seven additional benchmarks, including MVBench [9] as you suggested. As for ApolloBench [16], the benchmark is not publicly available at this time. We will evaluate our model on it when it becomes open-sourced.
>
> **Q4: “More details and visualizations of the dataset”**
>
> **A:** Thank you for the reminder. We will include additional details and visualizations of the dataset in the revised version.
>
> We illustrate the data distribution in Figure 3 and the dataset build-up pipeline in Figure 4. The videos span a wide range of categories, including travel, sports, education, animals, blogs, news, music, technology, comedy, entertainment, film, and gaming. The QA pairs are categorized into four reasoning types: **Temporal**, **Goal and Purpose**, **Spatial**, and **Plot and Narrative**. After thorough filtering and cleaning, the LongVideo-Reason dataset contains **104K** QA pairs with explicit reasoning annotations. Representative visual examples are provided in Figures 10 and 11.
>
> In addition, we further collected 2,000 additional 4K-resolution videos spanning scenarios such as **autonomous driving, video games, household robotics,** and **wildlife**. We employed VILA-HD [17] to generate object bounding boxes in video frames and constructed spatial reasoning Q&A pairs based on these boxes and corresponding captions.
>
> Additionally, the dataset contains a balanced mix of **multiple-choice** and **open-ended questions**, enhancing its versatility. To facilitate future research in video reasoning, we will release the dataset and the complete dataset construction pipeline to the community. We will also add more visualizations of the dataset in the future version of the paper.
>
> ### **References**
>
> [1] Video-MME: The First-Ever Comprehensive Evaluation Benchmark of Multi-modal LLMs in Video Analysis. CVPR 2025
>
> [2] LongVILA: Scaling Long-Context Visual Language Models for Long Videos. ICLR 2025
>
> [3] ActivityNet-QA: A Dataset for Understanding Complex Web Videos via Question Answering. AAAI 2019
>
> [4] Perception Test: A Diagnostic Benchmark for Multimodal Video Models. NeurIPS 2023
>
> [5] Needle In A Video Haystack: A Scalable Synthetic Framework for Benchmarking Video MLLMs. ICLR 2025
>
> [6] LongVideoBench: A Benchmark for Long-context Interleaved Video-Language Understanding. NeurIPS 2024
>
> [7] EgoSchema: A Diagnostic Benchmark for Very Long-form Video Language Understanding. NeurIPS 2023
>
> [8] NExT-QA: Next Phase of Question-Answering to Explaining Temporal Actions. CVPR 2021
>
> [9] MVBench: A Comprehensive Multi-modal Video Understanding Benchmark. CVPR 2024
>
> [10] Video-LLaVA: Learning United Visual Representation by Alignment Before Projection. EMNLP 2024
>
> [11] ShareGPT4Video: Improving Video Understanding and Generation with Better Captions. NeurIPS 2024
>
> [12] VideoLLaMA 2: Advancing Spatial-Temporal Modeling and Audio Understanding in Video-LLMs. ArXiv 2024
>
> [13] Kangaroo: A Powerful Video-Language Model Supporting Long-context Video Input. ArXiv 2024
>
> [14] LLaVA-OneVision: Easy Visual Task Transfer. Trans. Mach. Learn. Res. 2025
>
> [15] Video-R1: Reinforcing Video Reasoning in MLLMs. ArXiv 2025
>
> [16] Apollo: An Exploration of Video Understanding in Large Multimodal Models. CVPR 2025
>
> [17] Scaling Vision Pre-Training to 4K Resolution. CVPR 2025

---

> > ### Comment · Reviewer_QtJd · 2025-08-02
> >
> > Appreciate the comprehensive rebuttal, I will raise my rating.

---

> > > ### Author Response · Authors · 2025-08-03
> > >
> > > We sincerely appreciate your kind response and your recognition of our work, which truly means a lot to us.

---

### Official Review · Reviewer_HSHu · 2025-07-04

**Clarity:** 2
**Significance:** 2
**Originality:** 3
**Rating:** 4
**Confidence:** 4

**Summary:**

This paper introduces LongVILA-R1, a framework that enhances vision-language models (VLMs) for long video reasoning through reinforcement learning. The authors address challenges in long video understanding by developing three core components: (1) LongVideo-Reason dataset with 52K long video QA pairs with reasoning annotations, (2) a two-stage training pipeline combining chain-of-thought supervised fine-tuning (CoT-SFT) with reinforcement learning (RL), and (3) Multi-modal Reinforcement Sequence Parallelism (MR-SP) system for efficient long video RL training. The method scales from 16 to 512 frames and demonstrates competitive performance on VideoMME and their proposed LongVideo-Reason-eval benchmark.

**Questions:**

-	Comprehensive Benchmarking: Could the authors provide evaluation results on additional established long video benchmarks (such as EgoSchema, ActivityNet-QA, and LongVideoBench) to better demonstrate the generalizability and robustness of the reasoning improvements?
-	Cost-Benefit Analysis: Given the substantial computational investment required for the RL training pipeline, could the authors provide a more detailed analysis of the performance improvements achieved by their method compared to baseline models? Specifically, how do the gains justify the additional computational overhead and training complexity?

**Ethical Concerns:**

["NO or VERY MINOR ethics concerns only"]

**Final Justification:**

The authors have provided a comprehensive rebuttal addressing key concerns in my review. Hence, I decided to raise my rating accordingly.

**Limitations:**

Yes

**Paper Formatting Concerns:**

N//A

**Quality:**

2

**Strengths And Weaknesses:**

***Strengths***

-	The paper is generally well-written with clear explanations of the technical contributions.
-	The paper presents a well-structured approach to long video reasoning with clear methodology. The MR-SP system design is technically sound, achieving 9.2× speedup for long video RL training.
-	The dataset represents a valuable contribution as a resource for training reasoning models on video content.

***Weaknesses***

-	Missing Key Baseline: The evaluation lacks comparison with LongVILA, an important baseline for this work.
-	Limited Performance Gains: The performance improvements over existing baselines are modest and may not justify the substantial computational investment. Specifically, compared to the baseline LongVILA paper [1], LongVILA-7B achieved 60.1 (without subtitles) and 65.1 (with subtitles) on VideoMME with 256 frames, while LongVILA-R1-7B achieves only marginally better results at 60.3 (without subtitles) and 65.9 (with subtitles). Given the extensive additional training (CoT-SFT + RL) and computational resources required, these improvements appear insufficient.
-	Limited Evaluation Scope: The evaluation relies primarily on VideoMME and the authors' proprietary LongVideo-Reason-eval benchmark. More comprehensive evaluation across established long video benchmarks (such as EgoSchema, ActivityNet-QA, and LongVideoBench) would better validate the claimed reasoning improvements.

[1] Chen, Yukang, et al. "Longvila: Scaling long-context visual language models for long videos.", ICLR2025.

---

> ### Author Rebuttal · Authors · 2025-07-31
>
> ## **General Response**
>
> We sincerely appreciate your detailed and constructive feedback. Before addressing each concern individually, we first to highlight the major updates as follows.
>
> 1. **Larger Dataset:** We double the size of our dataset, LongVideo-Reason, from 52K to 104K reasoning data, with carefully filtering and data cleaning. Moreover, we include both multi-choice and open-ended QAs, making it more general.
> 2. **VideoMME Update**: Thanks to the updated dataset, our model achieved strong performance, e.g., **65.1% / 71.1%** on VideoMME [1] without and with subtitles, outperforming the LongVILA [2] baseline, which achieved 60.1% / 65.1% on VideoMME [1].
> 3. **Expanded Evaluation**: We have included **7 additional video benchmarks** for evaluation: ActivityNet-QA [3], PerceptionTest [4], VNBench [5],  LongVideoBench [6], EgoSchema [7],  NExT-QA [8], and MVBench [9]. Alongside VideoMME [1], we now compare LongVILA-R1 against LongVILA [2] baseline, and other methods [10, 11, 12, 13 ,14, 15] on a total of **8 benchmarks**, demonstrating consistently strong performance. Results are show in the table below.
> 4. **Extended Inputs**: LongVILA-R1-7B is capable of processing **2,048** video frames per video, with customizable FPS settings.
>
> | Model | VideoMME (w/o sub) | VideoMME (w/ sub) | ActivityNet-QA | PerceptionTest | VNBench | LongVideoBench | EgoSchema | NExT-QA | MVBench |
> | --- | --- | --- | --- | --- | --- | --- | --- | --- | --- |
> | VIdeo-LLaVA-7B | 39.9 | 41.6 | 45.3 | - | 12.4 | 37.6 | 38.4 | - | 43.5 |
> | ShareGPT4Video-8B | 39.9 | 43.6 | 50.8 | - | - | 41.8 | - | - | 51.2 |
> | VideoLLaMA2.1-7B | 54.9 | 56.4 | 53.0 | 54.9 | - | - | 53.1 | - | 57.3 |
> | Kangaroo-8B | 56.0 | 57.6 | - | - | - | 54.8 | 62.7 | - | 61.1 |
> | LLaVA-OV-7B | 58.2 | 61.5 | 56.7 | 57.1 | 51.8 | 56.4 | 60.1 | 79.4 | 56.7 |
> | Video-R1-7B | 61.4 | - | - | - | - | - | - | - | 64.8 |
> | LongVILA-7B | 60.1 | 65.1 | 59.5 | 58.1 | 63.0 | 57.1 | 67.7 | 80.7 | 67.1 |
> | LongVILA-R1-7B | **65.1** | **71.1** | **64.8** | **68.9** | **75.5** | **58.0** | **68.9** | **81.5** | **68.2** |
>
> ## **Detailed Response for Each Question**
>
> We provide detailed responses to each of the concerns below.
>
> **Q1: “Missing Key Baseline, Limited Performance Gains, Limited Evaluation Scope”**
>
> **A:** As mentioned in the **General Response**, we have evaluated LongVILA-R1-7B on eight benchmarks, including the three you suggested: ActivityNet-QA[3], LongVideoBench [6], and EgoSchema[7], and compared to the LongVILA-7B baseline. Across all benchmarks, LongVILA-R1-7B consistently outperforms LongVILA-7B  [9], achieving 65.1% and 71.1% on VideoMME [1], with **+5.0% and +6.0%** improvements over LongVILA-7B [2].
>
> **Q2: “Cost-Benefit Analysis”**
>
> **A**: Thank you for your constructive suggestion. We justify the performance gains from LongVILA-7B to LongVILA-R1-7B on VideoMME [1], along with the associated training overhead on H100.
> Both CoT SFT and RL fine-tuning contribute to meaningful improvements, from 65.1 → 66.9 → 71.1 (with subtitles). While CoT SFT and RL require 82 GPU-days and 224 GPU-days, the performance gains are non-trivial.
>
> In addition, as shown in the table in **General Response**, LongVILA-R1-7B achieves notable improvements over LongVILA-7B on several other benchmarks—for example, from **59.5% to 64.8% on ActivityNet-QA** [3], **58.1% to 68.9% on PerceptionTest** [4], and **63.0% to 75.5% on VNBench** [5].
>
> | Model | VideoMME (w/o sub) | VideoMME (w/ sub) | Overhead |
> | --- | --- | --- | --- |
> | Baseline | 60.1 | 65.1 | - |
> | +CoT SFT | 61.4 | 66.9 | 82 GPU days |
> | +RL | 65.1 | 71.1 | 224 GPU days |
>
> ### **References**
>
> [1] Video-MME: The First-Ever Comprehensive Evaluation Benchmark of Multi-modal LLMs in Video Analysis. CVPR 2025
>
> [2] LongVILA: Scaling Long-Context Visual Language Models for Long Videos. ICLR 2025
>
> [3] ActivityNet-QA: A Dataset for Understanding Complex Web Videos via Question Answering. AAAI 2019
>
> [4] Perception Test: A Diagnostic Benchmark for Multimodal Video Models. NeurIPS 2023
>
> [5] Needle In A Video Haystack: A Scalable Synthetic Framework for Benchmarking Video MLLMs. ICLR 2025
>
> [6] LongVideoBench: A Benchmark for Long-context Interleaved Video-Language Understanding. NeurIPS 2024
>
> [7] EgoSchema: A Diagnostic Benchmark for Very Long-form Video Language Understanding. NeurIPS 2023
>
> [8] NExT-QA: Next Phase of Question-Answering to Explaining Temporal Actions. CVPR 2021
>
> [9] MVBench: A Comprehensive Multi-modal Video Understanding Benchmark. CVPR 2024
>
> [10] Video-LLaVA: Learning United Visual Representation by Alignment Before Projection. EMNLP 2024
>
> [11] ShareGPT4Video: Improving Video Understanding and Generation with Better Captions. NeurIPS 2024
>
> [12] VideoLLaMA 2: Advancing Spatial-Temporal Modeling and Audio Understanding in Video-LLMs. ArXiv 2024
>
> [13] Kangaroo: A Powerful Video-Language Model Supporting Long-context Video Input. ArXiv 2024
>
> [14] LLaVA-OneVision: Easy Visual Task Transfer. Trans. Mach. Learn. Res. 2025
>
> [15] Video-R1: Reinforcing Video Reasoning in MLLMs. ArXiv 2025

---

> ### Author Response · Authors · 2025-08-05
> **Follow-up on Rebuttal for Paper 8328**
>
> Dear Reviewer HSHu,
>
> Thank you once again for your constructive and insightful review, which has significantly helped improve the quality of our work.
>
> Following your valuable suggestions, we have revised our paper and submitted a detailed rebuttal. In summary:
>
> 1. We conducted **comprehensive comparisons** across 8 benchmarks, including the key baseline LongVILA, and achieved non-trivial performance improvements (see **General Response & Q1**).
> 2. We provided a thorough **cost-benefit analysis**, showing that the training cost is reasonable and relative to the performance gains (see **Q2**).
>
> We hope these updates have addressed your concerns. As the discussion period is drawing to a close, we would greatly appreciate it if you could let us know whether our response has resolved your questions.
>
> Thank you again for your time and consideration.
>
> Sincerely,
>
> Authors of Paper 8328

---

> ### Author Response · Authors · 2025-08-06
> **Kind Reminder Regarding Rebuttal Feedback**
>
> Dear Reviewer HSHu,
>
> We sincerely appreciate your thoughtful and constructive review, which has greatly helped improve the quality of our work.
>
> Following your valuable feedback, we have carefully revised the paper and submitted a detailed rebuttal. As the discussion phase is coming to an end, we would greatly appreciate it if you could kindly let us know whether our response has addressed your concerns.
>
> Thank you again for your time and consideration.
>
> Sincerely,
>
> Authors of Paper 8328

---

> ### Author Response · Authors · 2025-08-08
> **Gentle Reminder Regarding Rebuttal Feedback**
>
> Dear Reviewer HSHu,
>
> We hope this message finds you well.
>
> Following your valuable review of our paper, we have carefully revised the paper and submitted a detailed rebuttal addressing your concerns. As the discussion period will end in just over one day, we would greatly appreciate it if you could let us know whether our responses have resolved your concerns.
>
> Thank you again for your time and contribution to the review process.
>
> Sincerely,
>
> Authors of Paper 8328

---

> ### Author Response · Authors · 2025-08-09
> **Final Reminder Regarding Rebuttal Feedback**
>
> Dear Reviewer HSHu,
>
> I hope this message finds you well.
>
> Following your valuable review of our paper, we have carefully addressed all your concerns in our rebuttal. Your review has been very helpful in improving our paper, and we have revised the paper based on your comments.
>
> As the discussion period will conclude in **about only 10 hours**, we would be very grateful if you could share your thoughts on our responses before the deadline. Your feedback would be invaluable in ensuring a complete and balanced discussion.
>
> Thank you again for your time and effort in reviewing our work.
>
> Sincerely,
>
> Authors of Paper 8328

---

> ### Comment · Area_Chair_QddH · 2025-08-09
> **Please engage in discussion**
>
> Dear Reviewer,
>
> The authors have responded to your reviews. Does it change your opinion? Please kindly engage in discussion and provide any feedback or questions if you may have. Thanks!
>
> AC

---

### Note · Authors · 2025-08-12

Dear Area Chairs and Reviewers,

We sincerely thank the Area Chairs and all reviewers for your time, effort, and thoughtful feedback. The detailed and constructive comments have been very helpful in improving our paper, and we have revised the paper accordingly.

We would like to provide an additional clarification summarizing the key strengths of this paper, as outlined below.

**Key strengths**:

1. **Valuable Dataset** – LongVideo-Reason is diverse and valuable to the community, containing 104K QAs with reasoning. [Reviewer HSHu, QtJd, oP4r, and XSh3]
2. **Training System** – Multi-modal Reinforcement Sequence Parallelism (MR-SP) is effective and improves efficiency for long video reasoning. [Reviewer HSHu, oP4r, and XSh3]
3. **Strong Results** – LongVILA-R1 achieves competitive performance, e.g., **65.1** and **71.1** on VideoMME without and with subtitles. [Reviewer QtJd, and oP4r]

Although Reviewer HSHu did not engage in the author–reviewer discussion, we are grateful for their valuable comments and would like to clarify how we addressed their concerns, with detailed evidence, as outlined below.

(1) “Missing Baseline / Limited Gains / Limited Scope“ - Addressed through comparing LongVILA-R1 with LongVILA on 8 benchmarks, showing notable improvements (e.g., **from 65.1 to 71.1** on VideoMME with subtitles). These concerns overlap with those from other reviewers [Reviewer XSh3 and QtJd], who have acknowledged them as resolved.

(2) “Cost-Benefit Analysis“ - Provided a detailed analysis of reasonable training costs and performance gains (e.g., **from 58.1% to 68.9%** on PerceptionTest). **The training cost (128 GPUs over 2.4 days) is well worth it, as this work is the first to enable reasoning on such large-scale and extra-long videos while achieving substantial performance gains.** Furthermore, LongVILA-R1 maintains the same architecture and model size as the LongVILA baseline, incurring **no extra overhead during inference**.

Finally, we will **open-source all relevant materials**, including the model, training data, benchmark, data generation pipeline, and the MR-SP system, to benefit the community.

Sincerely,

Authors of Paper 8328

---

### Decision · Program_Chairs · 2025-09-17

**Decision:**

Accept (poster)

**Comment:**

Final rating:  4: Borderline Accept/ 5. Accept/ 4: Borderline Accept/ 4: Borderline Accept. The paper introduces LongVILA-R1, a long-video reasoning framework combining a 52K LongVideo-Reason dataset, a two-stage CoT-SFT→RL pipeline, and MR-SP training (sequence parallelism + vLLM with cached embeddings). LongVILA-R1-7B shows strong VideoMME results, surpasses Video-R1-7B by 5.2%, matches Gemini-1.5-Pro on the authors’ eval, achieves up to 9.2× RL speedups, and scales from 16 to 512 frames.

Reviewers commend the valuable LongVideo-Reason dataset and a polished training pipeline with strong results, but initially flagged: no LongVILA baseline and only marginal VideoMME gains despite costly CoT-SFT+RL; narrow evaluation (VideoMME + in-house), limited diversity (Shot2Story-based) and modest novelty (SFT→GRPO); incomplete SOTA/fairness comparisons (fine-tuned on own eval vs zero-shot rivals); and scalability/robustness limits (≤512 frames; unclear at high/variable frame rates).

After rebuttal, reviewers said concerns were addressed and leaned toward acceptance. The ACs concur—citing the dataset’s value (104K reasoning QA pairs), efficient MR-SP training, and competitive VideoMME scores (65.1/71.1 w/o/with subtitles)—and recommend acceptance.